# Online Min-max Problems with Non-convexity and Non-stationarity

**Yu Huang**[*]                                            *y-huang20@mails.tsinghua.edu.cn*
*Institute for Interdisciplinary Information Sciences*
*Tsinghua University*
**Yuan Cheng**[*]                                            *cy16@mail.ustc.edu.cn*
*University of Science and Technology of China*
**Yingbin Liang**                                            *liang.889@osu.edu*
*Department of Electrical and Computer Engineering*
*The Ohio State University*
**Longbo Huang**[†]                                            *longbohuang@tsinghua.edu.cn*
*Institute for Interdisciplinary Information Sciences*
*Tsinghua University*

**Reviewed on OpenReview:** *https://openreview.net/forum?id=TdzQtbLeVw*

## Abstract

Online min-max optimization has recently gained considerable interest due to its rich applications to game theory, multi-agent reinforcement learning, online robust learning, etc. Theoretical understanding in this field has been mainly focused on convex-concave settings. Online min-max optimization with nonconvex geometries, which captures various online deep learning problems, has yet been studied so far. In this paper, we make the first effort and investigate online nonconvex-strongly-concave min-max optimization in the nonstationary environment. We first introduce a natural notion of local Nash equilibrium (NE)-regret, and then propose a novel algorithm coined TSODA to achieve the optimal regret. We further generalize our study to the setting with stochastic first-order feedback, and show that a variation of TSODA can also achieve the same optimal regret in expectation. Our theoretical results and the superior performance of the proposed method are further validated by empirical experiments. To our best knowledge, this is the first exploration of efficient online nonconvex min-max optimization.

## 1 Introduction

Online optimization (Cesa-Bianchi & Lugosi, 2006) is a powerful paradigm for modeling many applications that require decision making based on information available sequentially. Specially, at each time instance, an online player needs to make a decision based on the history information, and then receives a feedback (which can be a possibly adversarial and nonstationary reward or loss value) that may be used in the future. There have been extensive studies in this field for various scenarios, such as online convex optimization (Shalev-Shwartz, 2012; Hazan et al., 2016), online bilevel optimization (Tarzanagh & Balzano, 2022), online federated learning (Chen et al., 2020), etc. Recently, the online min-max (i.e., saddle point) problem has gained considerable interest due to its broad applications in game theory (Roy et al., 2019; Zhang et al., 2022), multi-agent reinforcement learning (Buşoniu et al., 2010; Zhang et al., 2021), online robust learning (Gabrel et al., 2014; Ben-Tal et al., 2015), to name a few.

On the theoretical side, a line of works have explored provably efficient algorithms for online min-max optimization. Specifically, Cardoso et al. (2019); Fiez et al. (2021); Immorlica et al. (2019); Zhang et al. (2022) considered the zero-sum matrix games where the online objective function takes a bilinear form.

---

[*]Equal Contributions
[†]Corresponding author

Rivera et al. (2018); Roy et al. (2019) studied a more general online min-max problem, where the objective is strongly-convex and strongly-concave. Noarov et al. (2021) focused on multi-objective online min-max games, where the reward is convex-concave in each coordinate.

Despite many efforts so far, existing literature on online min-max optimization has mainly focused on online convex-concave problems and did not take **nonconvexity** into consideration. However, in practice, nonconvexity occurs very often in online min-max problems, particularly those that apply deep neural networks (DNNs) for decision making. For instance, in the time-varying two-player zero-sum stochastic games (Mertens & Neyman, 1981; Roy et al., 2019; Zhang et al., 2022), where the payoffs change with time, the policies are modeled by DNNs with strong regularization, and hence the online objective function is nonconvex-strongly-concave.

Motivated by the aforementioned practical problems, the goal of this paper is to take the first step towards exploring the **online nonconvex-strongly-concave min-max** problem with dynamic (and hence nonstationary) loss functions. Due to the nonconvexity and nonstationarity nature of the problem, two new challenges arise as we explain below.

First, *how to define an appropriate notion of regret for the nonstationary environment under the online nonconvex setting?* The standard notion of Nash Equilibrium (NE)-regret, e.g., Rivera et al. (2018) for online convex-concave problems, which quantifies the difference between the cumulative loss of players and the min-max value of the cumulative payoff loss, is highly unreasonable for nonconvex-concave setting, since the min-max comparator is intractable for a nonconvex-concave function. Hence, new surrogate for regret is in demand.

Second, *with a desirable notion of regret, how to design efficient algorithms?* A natural strategy to handle the nonstationarity is that at each round, the decision maker first learns a good enough decision based on history losses and then applies it to the adversarial loss of current round. Two key difficulties will arise during this process. First, how to identify a good decision? In nonconvex min-max problems, a good decision usually refers to a stationary point. The standard definition of a stationary point involves an optimization oracle, which is unknown to the decision maker. Thus the decision maker needs to find a surrogate to identify a near stationary point at each round. Second, when applying the decision based on history information to the adversarial loss, mismatch errors arise due to the variability of the environment, which motivates the need for nonstationarity measures.

## 1.1 Our contributions

In this paper, we handle the aforementioned challenges by introducing a new regret measure and developing efficient algorithms for online nonconvex min-max problem with optimal regret guarantees. The main contributions are highlighted below.

- We first introduce a novel notion of dynamic regret for online nonconvex-strongly-concave min-max problem, called **local Nash equilibrium (NE)-regret**, which jointly captures the nonconvexity, nonstationarity, and min-max nature of our problem.

- Based on the regret notion, we propose an efficient online min-max optimization algorithm, named Time-Smoothed Online gradient Descent Ascent (TSODA). The main idea underlying TSODA is to output a near-stationary point at each round by performing two-timescale gradient descent ascent and utilizing a specially designed stop criterion component.

- We show that the local NE-regret of TSODA scales as $O(\frac{T}{w^2})$ with a iteration complexity of $O(Tw)$, where $T$ represents the total number of rounds and $w$ denotes the size of the sliding window used to define local NE-regret. Such result matches the $\Omega(\frac{T}{w^2})$ regret lower bound and the order of iteration complexity of $O(Tw)$ provided in Hazan et al. (2017a) for online minimization (where we set the maximization to be over a singleton). Thus, TSODA achieves the optimal performance for online nonconvex-strongly-concave min-max optimization.

- We further generalize our study to the setting with stochastic first-order feedback and show that a variation of TSODA can also achieve a regret of $O(\frac{T}{w^2})$.

- We verify our theoretical results and demonstrate the effectiveness of our algorithm through several empirical experiments on real-world datasets.

To our best knowledge, this is the first study on online nonconvex min-max optimization with theoretical characterization of the regret performance.

## 1.2 Related Work

**Online Min-max Optimization.** Recently, online min-max optimization, also known as online saddle-point game, has emerged as an interesting optimization framework, and has been studied under various settings. More specifically, the zero-sum matrix game considers the special case that the function is bilinear with a payoff matrix $\mathbf{A}_t$, where the objective function is given by $f_t(\mathbf{x}, \mathbf{y}) = \mathbf{x}^\top \mathbf{A}_t \mathbf{y}$. Several works, for example, Cardoso et al. (2019); Fiez et al. (2021); Immorlica et al. (2019); Zhang et al. (2022) proposed and analyzed algorithms with respect to different notions of regret. For more general objective functions, Rivera et al. (2018); Roy et al. (2019) studied the case where the loss function $f_t$ is strongly-convex-strongly-concave. Very recently, Noarov et al. (2021) formulated a general multi-objective framework, where the goal is to minimize the maximum coordinate of the cumulative vector-valued loss with convex-concave function in every coordinate. We emphasize that all of the above studies did not consider nonconvexity in their objective functions, which is the focus of our study here.

**Online Nonconvex Optimization.** As online nonconvex optimization is an active research area, various works have taken different approaches to handle the nonconvexity. Assuming access to an offline nonconvex optimization oracle to approximate minimizers of perturbed nonconvex functions, Suggala & Netrapalli (2020); Agarwal et al. (2019) studied the performance of "follow the perturbed leader" (FTPL) algorithm (Kalai & Vempala, 2005), and their regrets are all static regret. Further, Hazan et al. (2017a); Hallak et al. (2021); Aydore et al. (2019) considered online nonconvex problems under nonstationary environments, and utilized sliding windows method with window size $w$. They proposed different notions of dynamic regrets and algorithms, and achieved an order of $O(\frac{T}{w^2})$ according regret notions. Additionally, Héliou et al. (2020) studied online nonconvex optimization with imperfect feedback. Except first-order optimization, Héliou et al. (2020); Roy et al. (2022) considered zeroth-order online nonconvex optimization and Lesage-Landry et al. (2020) studied second-order online nonconvex optimization.

**Offline Min-max Optimization.** There is a rich literature that studies a diverse set of algorithms for min-max optimization with nonconvexity in the offline setting. We next describe only those studies highly relevant to our study here. One celebrated approach is the nested-loop type algorithm (Rafique et al., 2021; Nouiehed et al., 2019; Thekumparampil et al., 2019; Kong & Monteiro, 2021), where the outer loop can be treated as an inexact gradient descent on a nonconvex function while the inner-loop aims to find an approximate solution to the maximization problem (see Lin et al. (2020a) and references therein for a good collection of such studies). Another approach, manifesting in the recent works of Lu et al. (2020) and Lin et al. (2020a) considers less complicated single-loop structures. Specifically, the two-timescale GDA analyzed in Lin et al. (2020a) is closest to the implementation at each round of our proposed TSODA method. But it is not straightforward to generalize the design to the online setting, and our analysis of the new local NE-regret for online optimization is also very different from such a offline min-max problem.

## 1.3 Notations

$[T] \triangleq \{1, \dots, T\}$. We use bold lower-case letters to denote vectors as in $\mathbf{x}, \mathbf{y}$, and denote its $\ell_2$-norm as $\|\cdot\|$. We use calligraphic upper case letters to denote sets as in $\mathcal{Y}$, and use the notation $\mathcal{P}_\mathcal{Y}$ to denote projections onto the set. For a differentiable function $\Phi(\cdot) : \mathbb{R}^m \to \mathbb{R}$, we let $\nabla \Phi(\mathbf{x})$ denote the gradient of $\Phi$ at $\mathbf{x}$. For a function $f(\cdot, \cdot) : \mathbb{R}^m \times \mathcal{Y} \to \mathbb{R}$ of two variables, $\nabla_\mathbf{x} f(\mathbf{x}, \mathbf{y})$ (or $\nabla_\mathbf{y} f(\mathbf{x}, \mathbf{y})$) denotes the partial gradient of $f$ with respect to the first variable (or the second variable) at the point $(\mathbf{x}, \mathbf{y})$. We also use $\nabla f(\mathbf{x}, \mathbf{y})$ to denote the full gradient at $(\mathbf{x}, \mathbf{y})$ where $\nabla f(\mathbf{x}, \mathbf{y}) = (\nabla_\mathbf{x} f(\mathbf{x}, \mathbf{y}), \nabla_\mathbf{y} f(\mathbf{x}, \mathbf{y}))$. Finally, we use the notation $O(\cdot)$ and $\Omega(\cdot)$ to hide constant factors which are independent of problem parameters.

## 2 Problem Setup

We consider solving the following online min-max (i.e., saddle-point) problem:

$$\min_{\mathbf{x} \in \mathbb{R}^m} \max_{\mathbf{y} \in \mathcal{Y}} f_t(\mathbf{x}, \mathbf{y}), \quad t \in [T] \tag{1}$$

where $f_t : \mathbb{R}^m \times \mathbb{R}^n \to \mathbb{R}$ is generally *nonconvex* in $\mathbf{x}$ but *concave* in $\mathbf{y}$ and where $\mathcal{Y}$ is a convex set. Such choice of unbounded $\mathbf{x}$ and bounded $\mathbf{y}$ is commonly used in existing analysis for nonconvex-concave problems (Lin et al., 2020a; Li et al., 2022; Yang et al., 2022). In our work, such an assumption brings the technical convenience by allowing us to control $\delta_{t,w}^0 = \left\| \mathbf{y}_{t,w}^\star \left( \mathbf{x}_t^0 \right) - \mathbf{y}_t^0 \right\|^2$ at round $t = 1$ (see Section 6.2 for details). At each round $t \in [T]$, the environment first incurs a loss function $f_t$. Without knowing the knowledge of $f_t$, the $\mathbf{x}$-learner and $\mathbf{y}$-learner are tasked with predicting $\mathbf{x}_t$ and $\mathbf{y}_t$ respectively to solve eq. (1) based on loss functions up to round $t - 1$, i.e., $\{f_i\}_{i=1}^{t-1}$. The learners then observe the function $f_t(\cdot)$ and suffer a loss of $f_t(\mathbf{x}_t, \mathbf{y}_t)$.

The following regularity assumptions for $f_t$ are made throughout the entire paper:

**Assumption 1** (Smoothness). *$f_t$ is $\ell$-smooth $\forall t \in [T]$, i.e., $\forall (\mathbf{x}, \mathbf{y}), (\mathbf{x}', \mathbf{y}')$, it holds that $\|\nabla f_t(\mathbf{x}, \mathbf{y}) - \nabla f_t(\mathbf{x}', \mathbf{y}')\| \leq \ell \|(\mathbf{x}, \mathbf{y}) - (\mathbf{x}', \mathbf{y}')\|$.*

**Assumption 2** (Strong Concavity). *The function $f_t(\mathbf{x}, \cdot)$ is $\mu$-strongly concave $\forall t \in [T]$, i.e., given $\mathbf{x} \in \mathbb{R}^m$, $\forall \mathbf{y}, \mathbf{y}'$, it holds that $f_t(\mathbf{x}, \mathbf{y}) \leq f_t(\mathbf{x}, \mathbf{y}') + \langle \nabla_\mathbf{y} f_t(\mathbf{x}, \mathbf{y}'), \mathbf{y} - \mathbf{y}' \rangle - \frac{\mu}{2} \|\mathbf{y} - \mathbf{y}'\|^2$.*

**Assumption 3** (Boundness). *The set $\mathcal{Y}$ is a convex and bounded set with diameter $D \geq 0$. There exists $M > 0$, s.t. $|f_t(\mathbf{x}, \mathbf{y})| \leq M$, $\forall t \in [T], \mathbf{x} \in \mathbb{R}^m, \mathbf{y} \in \mathcal{Y}$.*

The above assumptions are standard in the literature of online learning (Hazan et al., 2017b) and min-max optimization (Lin et al., 2020a;b). While our analysis primarily focuses on the nonconvexity of $\mathbf{x}$, it is worth mentioning that our approach can be extended to the nonconvex-concave setting by employing a weaker condition for $\mathbf{y}$, as discussed in Lin et al. (2020a).

When the loss $f_t$ is fixed for all $t$, our framework specializes to the standard nonconvex-strongly-concave min-max optimization (Lin et al., 2020a;b). Putting into the context of online min-max optimization, our formulation is similar to those in Roy et al. (2019); Rivera et al. (2018); Zhang et al. (2022), where they studied only the case where $f_t$ is convex-concave. However, their standard regret minimization and equilibrium computation will be computationally infeasible for general nonconvex-strongly-concave losses. Next, we provide a motivating example for the online nonconvex-concave min-max optimization problem that we study here.

**Motivating Application.** Generative Adversarial Networks (GANs) (Goodfellow et al., 2014) are a popular machine learning model, in which a generator network $G_\mathbf{x}(\cdot)$ playes against a discriminator network $D_\mathbf{y}(\cdot)$ via a min-max formulation given by:

$$\min_\mathbf{x} \max_\mathbf{y} M(\mathbf{x}, \mathbf{y}) = \frac{1}{2} \mathbb{E}_{\mathcal{P} \sim p_{\text{data}}} \log D_\mathbf{y}(\mathcal{P}) + \frac{1}{2} \mathbb{E}_{\mathcal{Q} \sim p_{\text{noise}}} \log \left(1 - D_\mathbf{y}\left(G_\mathbf{x}(\mathcal{Q})\right)\right).$$

In practice, GANs are commonly trained with deep architectures, where both the discriminator and the generator are deep neural networks, making GANs hard to optimize and analyze. To address such a challenge, it is theoretically sound to consider an intermediate setting also arising in many real-world scenarios, called GANs with semi-shallow architectures (Grnarova et al., 2017; Moghadam et al., 2021), where the generator $G_\mathbf{x}(\cdot)$ is any arbitrary deep neural network and the discriminator $D_\mathbf{y}(\cdot)$ consists of a single layer network. Such an architecture naturally yields a nonconvex-concave game, i.e., $M(\mathbf{x}, \mathbf{y})$ is nonconvex in $\mathbf{x}$ and concave in $\mathbf{y}$. Furthermore, there is a growing demand for GANs to handle *time-varying* scenarios (Mogren, 2016; Esteban et al., 2017; Yoon et al., 2019), such as time-series data, where generated samples should preserve the temporal dynamics of the data. This requires the GAN's model parameters to be updated in real-time to adapt to the changes in the data distribution, which leads to an online setting where the objective function $M_t(\mathbf{x}, \mathbf{y})$ changes over time $t$. Therefore, by combining the above facts, the training of time-varying GANs can be captured by online nonconvex-concave min-max problems. Solving such a type of problems has the potential to advance the field of generative models, particularly in scenarios where the data distribution changes over time.

## 3 How to Measure the Performance?

### 3.1 Local Nash Equilibrium (NE)-Regret

We introduce a new definition of a local regret that suits online nonconvex-strongly-concave min-max problems. Our new metric is motivated by the online nonconvex optimization literature; see for example Hazan et al. (2017a); Hallak et al. (2021). Specifically, for each $t$, we first define the smoothed functions of $f_t$ over a sliding-window of size $w$ as:

$$F_{t,w}(\mathbf{x}, \mathbf{y}) \stackrel{\text{def}}{=} \frac{1}{w} \sum_{i=0}^{w-1} f_{t-i}(\mathbf{x}, \mathbf{y}). \tag{2}$$

For notation convenience, we treat $f_t(\mathbf{x}, \mathbf{y})$ as 0 for all $t < 0$. Moreover, since the averaging preserves strongly-convexity, which implies $F_{t,w}$ is strongly-concave in $\mathbf{y}$, the maximization problem $\max_{\mathbf{y} \in \mathcal{Y}} F_{t,w}(\mathbf{x}, \mathbf{y})$ can be solved efficiently. Then, we can naturally define the following function:

$$\Phi_{t,w}(\mathbf{x}) \stackrel{\text{def}}{=} \max_{\mathbf{y} \in \mathcal{Y}} F_{t,w}(\mathbf{x}, \mathbf{y}). \tag{3}$$

The overall goal of online min-max optimization can be viewed as online minimization over the above defined $\Phi_{t,w}(\cdot)$ function. Thus, we define the following regret metric with respect to $\Phi_{t,w}(\cdot)$.

**Definition 1** (Local Nash Equilibrium (NE)-Regret). *Let $f_t$ be a sequence of functions satisfying Assumption 1-3, with $\Phi_{t,w}(\cdot)$ defined in eq. (3). The w-local Nash Equilibrium (NE)-Regret is defined as:*

$$\mathfrak{R}_w^{NE}(T) \stackrel{\text{def}}{=} \sum_{t=1}^{T} \|\nabla \Phi_{t,w}(\mathbf{x}_t)\|^2. \tag{4}$$

$\nabla \Phi_{t,w}$ is well-defined since $\Phi_{t,w}$ is differentiable for nonconvex-strongly-concave min-max problem (Lin et al., 2020a). We next justify the above notion of the local NE-regret from three aspects.

**Why Norm of Gradient as Metric?** At each round $t$ of the nonconvex-concave min-max optimization problem, the objective function can be expressed as $\min_{\mathbf{x} \in \mathbb{R}^m} \Gamma_t(\mathbf{x})$, where $\Gamma_t(\cdot) = \max_{\mathbf{y} \in \mathcal{Y}} f_t(\cdot, \mathbf{y})$ is generally nonconvex, and hence finding the global minimum for $\Gamma_t(\mathbf{x})$ is NP hard. A common surrogate for the global minimum of $\Gamma_t$ in the offline nonconvex-strongly-concave min-max literature is the notion of $\epsilon$-stationary point (Lin et al., 2020a;b) for a differentiable $\Gamma_t$, i.e., an output $\mathbf{x}$ such that $\|\nabla \Gamma_t(\mathbf{x})\|^2 < \epsilon$. If $\epsilon = 0$, then $\mathbf{x}$ is a stationary point. Therefore, it is reasonable to leverage such a norm of gradient as the optimality criterion for online nonconvex-concave min-max optimization.

**Why Sliding-window Averaging?** The motivation behind the window averaging is two-fold: (*i*) $F_{t,w}$ and $\Phi_{t,w}$ represent the average performance during the window, which is widely adopted to handle noises and fluctuations when the environment and the loss function $f_t$ encounter mild perturbations and variations. For instance, when each loss function $f_t$ is an unbiased noisy realization of some $f$, the expected gradient norm of a randomly selected update inside the window is a standard measure in the nonconvex stochastic optimization literature (Bottou et al., 2018) and can reduce the variation caused by noises. Such smoothed notion is also a common practice in the field of online nonconvex optimization[1] (Hazan et al., 2017a; Hallak et al., 2021; Aydore et al., 2019; Zhuang et al., 2020). (*ii*) In practice, the average performance of a system is a typical and intuitive notion commonly used to evaluate real-world applications. Suppose a decision maker in a time-varying environment (with loss functions $f_t$) has only finite term memory $w$. Then she naturally wishes to find the best decision based on the entire finite term memory and will choose the average loss function $F_{t,w}$ and $\Phi_{t,w}$ as the performance metrics. As another example, if the environment varies in a periodic manner, such an average performance metric during a whole period is naturally adopted in time series forecasting problems.

**Why Capturing the Dynamic Nature?** It is desirable that the regret can capture how well the players adapt their actions to the best decision at *each round* if the environment is nonstationary and changes over time. In the well-studied online convex-concave setting, the notion of dynamic NE-regret (Roy et al., 2019; Zhang et al., 2022) is defined for this purpose, since its definition of $|\sum_{t=1}^{T} f_t(\mathbf{x}_t, \mathbf{y}_t) -$

---

[1]If we view $\mathcal{Y}$ to be singleton, the local NE-regret degenerates to local regret proposed in Hazan et al. (2017a).

$\sum_{t=1}^{T} \min_{\mathbf{x} \in \mathbb{R}^m} \max_{\mathbf{y} \in \mathcal{Y}} f_t(\mathbf{x}, \mathbf{y})|$ evaluates the gap to the min-max comparator at *each round* instead of the min-max solution of the sum of functions over all rounds. For the nonconvex min-max setting, the best min-max comparator at *each round* can be set as the stationary point of the window function $\Phi_{t,w}(\cdot)$, which has zero gradient. Hence, our local regret in eq. (4) can be interpreted as evaluating the gap between $\|\nabla \Phi_{t,w}(\mathbf{x}_t)\|^2$ and its comparator (which equals zero gradient) at *each round*, and thus implicitly captures the player's adaption to the dynamic setting. In the special strongly-convex-strongly-concave case, under some mild continuity conditions, a lower local NE-regret with $w = 1$ implies a lower dynamic NE-regret. We provide a concrete toy example in Appendix A to illustrate this relationship.

### 3.2 Variability of Environment

Intuitively, if the environment (and hence the loss function $f_t$) changes drastically over time, it will be hard to obtain meaningful guarantees efficiently. To handle this problem, dynamic (Roy et al., 2019; Zhang et al., 2022) or local (Hallak et al., 2021) regret serves as better performance metrics to take the changing environment into consideration. Such notions typically rely on certain nonstationarity measures of the environment in order to reflect how the system dynamics affects the performance. Therefore, in this subsection, we introduce such measures of variation for loss functions, which will be crucial in our analysis and capture the nonstationarity of our online min-max settings.

**Definition 2** (Variation of Sliding-window). *Let us denote* $\mathbf{y}_{t,w}^*(\mathbf{x}) = \arg \max_{\mathbf{y} \in \mathcal{Y}} F_{t,w}(\mathbf{x}, \mathbf{y})$. *We define the following two types of sliding-window variation:*

$$V_w^1[T] := \sum_{t=1}^{T} \sup_{\mathbf{x} \in \mathbb{R}^m} \|\nabla_{\mathbf{x}} f_t\left(\mathbf{x}, \mathbf{y}_{t,w}^*(\mathbf{x})\right) - \nabla_{\mathbf{x}} f_{t-w}\left(\mathbf{x}, \mathbf{y}_{t,w}^*(\mathbf{x})\right)\|^2, \tag{5}$$

$$V_w^2[T] := \sum_{t=1}^{T} \sup_{\mathbf{x} \in \mathbb{R}^m} \|\nabla_{\mathbf{y}} f_t\left(\mathbf{x}, \mathbf{y}_{t,w}^*(\mathbf{x})\right) - \nabla_{\mathbf{y}} f_{t-w}\left(\mathbf{x}, \mathbf{y}_{t-1,w}^*(\mathbf{x})\right)\|^2. \tag{6}$$

*Remark* 1. $V_w^1[T]$ primarily measures the drift of $f_t$ and $f_{t-w}$ in $\mathbf{x}$, considering that the $\mathbf{y}$-players for these models are determined by $\mathbf{x}$ through $\mathbf{y}_{t,w}^*(\cdot)$. On the other hand, $V_w^2[T]$ further quantifies the changes of the maximum players for $F_{t,w}(\mathbf{x}, \cdot)$ and $F_{t-1,w}(\mathbf{x}, \cdot)$, i.e. $\mathbf{y}_{t,w}^*$ and $\mathbf{y}_{t-1,w}^*$. Therefore, by considering both $V_w^1[T]$ and $V_w^2[T]$, we can jointly capture the variations in the environments of the online min-max problem.

*Remark* 2. Clearly, $V_w^1[T]$ and $V_w^2[T]$ are $O(T)$ if the gradients of $f_t$ are bounded and can be zero in the offline setting, i.e., $T = 1$. A key observation is that if the loss function encounters a periodic shift with certain period length of $w^*$, i.e., $f_{t+w^*} = f_t$, then for $w = w^*$ and $t \geq w$, we have $f_t = f_{t-w}$ and $\mathbf{y}_{t,w}^* = \mathbf{y}_{t-1,w}^*$, which is implied by the fact that $F_{t+1,w} = F_{t,w}$. As a consequence, for a well-tuned $w \ll T$, the sliding-window variations could be considerably smaller compared to $T$, especially $V_w^1[T] = V_w^2[T] = O(w)$ in the above case.

## 4 TSODA: Time-Smoothed Online Gradient Descent Ascent

In this section, we present our proposed method, named time-Smoothed Online gradient Descent Ascent (TSODA), for online nonconvex-strongly-concave problem, and we show that our approach is capable of efficiently achieving a favorable local NE-regret bound.

### 4.1 Proposed Algorithm

At the high-level, our algorithm plays following-the-leader iterates, aiming to find a suitable approximating stationary point at each round using two-timescale gradient descent ascent (GDA). At each round $t$, TSODA performs gradient descent over the variable $\mathbf{x}$ with the stepsize $\eta_{\mathbf{x}}$ and gradient ascent over the variable $\mathbf{y}$ with the stepsize $\eta_{\mathbf{y}}$ on function $F_{t,w}(\mathbf{x}, \mathbf{y})$ until the following Stop Condition 1 is satisfied. Then, TSODA observes the loss function $f_{t+1}$ to be used in the next round. The pseudocode of TSODA is summarized in Algorithm 1.

*Stop Condition* 1. The terminating condition for Algorithm 1 is:

$$\left((\frac{2\kappa}{\eta_{\mathbf{y}}} + \ell)(1 + \ell\eta_{\mathbf{y}})\right)^2 \|\mathbf{y}_{t+1} - \mathcal{P}_{\mathcal{Y}}\left(\mathbf{y}_{t+1} + \eta_{\mathbf{y}}\nabla_{\mathbf{y}} F_{t,w}\left(\mathbf{x}_{t+1}, \mathbf{y}_{t+1}\right)\right)\|^2 + \|\nabla_{\mathbf{x}} F_{t,w}(\mathbf{x}_{t+1}, \mathbf{y}_{t+1})\|^2 \leq \frac{\delta^2}{2w^2}. \tag{7}$$

---

**Algorithm 1 T**ime-**S**moothed **O**nline Gradient **D**escent **A**scent (TSODA)

---

**Input:** window size $w \geq 1$, stepsizes $(\eta_{\mathbf{x}}, \eta_{\mathbf{y}})$, tolerance $\delta > 0$
**Initialization:** $(\mathbf{x}_1, \mathbf{y}_1)$
 1: **for** $t = 1$ to $T$ **do**
 2:     Predict $(\mathbf{x}_t, \mathbf{y}_t)$. Observe the cost function $f_t : \mathbb{R}^m \times \mathbb{R}^n \to \mathbb{R}$
 3:     Set $(\mathbf{x}_{t+1}, \mathbf{y}_{t+1}) \leftarrow (\mathbf{x}_t, \mathbf{y}_t)$
 4:     **repeat**
 5:         $\mathbf{x}_{t+1} \leftarrow \mathbf{x}_{t+1} - \eta_{\mathbf{x}} \nabla_{\mathbf{x}} F_{t,w}(\mathbf{x}_{t+1}, \mathbf{y}_{t+1})$
 6:         $\mathbf{y}_{t+1} \leftarrow \mathcal{P}_{\mathcal{Y}}(\mathbf{y}_{t+1} + \eta_{\mathbf{y}} \nabla_{\mathbf{y}} F_{t,w}(\mathbf{x}_{t+1}, \mathbf{y}_{t+1}))$
 7:     **until** Equation (7) in Stop Condition 1 holds
 8: **end for**

---

**Discussions about Stopping Criterion.** Due to the online nature, the design of the stopping condition is to guarantee that the learner outputs a good $\mathbf{x}_{t+1}$ with small local regret at round $t$, i.e., $\|\nabla \Phi_{t,w}(\mathbf{x}_{t+1})\|^2$ is small enough. However, in contrast to general online nonconvex games (Hazan et al., 2017a), where the first-order information is available, we do not have direct access to the first-order oracle of $\Phi_{t,w}$. To circumvent this issue, we adopt the global error bound condition from the seminal paper (Drusvyatskiy & Lewis, 2018) to translate conditions on $\nabla \Phi_{t,w}(\mathbf{x}_{t+1})$ into restrictions on tractable $\nabla F_{t,w}$. Specifically, we prove that $\|\nabla \Phi_{t,w}(\mathbf{x}_{t+1})\|^2$ is upper bounded by the left-hand side (LHS) of inequality in Stop Condition 1 (see Lemma 6.1). Therefore, alternatively we can utilize the accessible information of $\nabla F_{t,w}$ to terminate the inner-loop iterations at time $t$.

**Last-iterate Guarantee.** At each round $t$, the stop condition will be triggered only when the local regret of last iteration is small enough. Such a *last-iterate* type guarantee is different by nature from existing offline nonconvex-strongly-concave min-max results (Lin et al., 2020b;a), which are only guaranteed to visit an $\epsilon$-stationary point within a certain number of iterations, i.e., where the return $\bar{\mathbf{x}}$ is uniformly drawn from previous iterations. Crucially, we will establish the total iteration bound (see Theorem 2) in the next subsection, which indicates that such last-iterate type outputs can be obtained efficiently. Furthermore, since the stopping criterion leads to stronger guarantee, our result is incomparable with former offline iteration complexity in the special case that $T = 1$.

### 4.2 Theoretical Guarantees

In this subsection, we provide the regret and computational complexity guarantees of our algorithm under local NE-regret and highlight several connections with the existing results from offline min-max optimization and online nonconvex problem.

**Theorem 1** (Local NE Regret Minimization)**.** *Let $\kappa = \ell/\mu$ denote the condition number. Under Assumptions 1-3, and letting the stepsizes be chosen as $\eta_{\mathbf{x}} = \Theta\left(1/\kappa^3 \ell\right)$ and $\eta_{\mathbf{y}} = \Theta(1/\ell)$, then Algorithm 1 enjoys the following local NE-regret bound:*

$$\Re_w^{NE}(T) = \sum_{t=1}^T \|\nabla \Phi_{t,w}(\mathbf{x}_t)\|^2 \leq \frac{3}{w^2}(T\delta^2 + \frac{(\kappa w)^2}{(w-1)^2} V_w^2[T] + V_w^1[T]).$$

**Theorem 2** (Iteration Bound)**.** *Let $\tau$ denote the total number of iterations incurred by Algorithm 1. Then $\tau$ can be upper bounded as:*

$$\tau \leq \frac{384\kappa^3 \ell M w T}{\delta^2} + 576\frac{\kappa^2 T}{\mu} + \frac{576 D^2 \kappa^3 \ell^2 w^2}{\delta^2} + 1152\frac{w^2 \kappa^5}{(w-1)^2 \delta^2} V_w^2[T].$$

*Furthermore, the number of first-order gradient calls is bounded by $O(w\tau)$.*

Theorems 1 and 2 together reveal the trade-off between the impact of sliding-window size $w$ on the regret and the computational complexity, where larger $w$ leads to smaller regret bound but incurs more gradient calls.

**Robustness of TSODA.** Our results in Theorems 1 and 2 are expressed in terms of variation measures $V_w^1[T]$ and $V_w^2[T]$ of the environment introduced in Section 3.2. If we make the same assumption similar to

that in Hazan et al. (2017a) that the gradient of $f_t$ is bounded, the above theorems provide a robust guarantee for TSODA; namely, no matter how the environment changes at each round, TSODA always ensures $O(\frac{T}{w^2})$ local NE-regret with $O(Tw)$ iterations since $V_w^1[T]$ and $V_w^2[T]$ are $O(T)$ by definitions. Therefore, the regret can be made sublinear in $T$ if $w$ is selected accordingly. Interestingly, depending on the degree of nonstationarity, TSODA is capable of achieving even smaller local NE-regret.Particularly, as we discussed in Remark 2, for the scenario that $f_t$ is periodic with period $w \ll T$, $V_w^1[T] = V_w^2[T] = O(w)$.

**Optimality of Regret Bound.** Note that the basic online nonconvex minimization problem can be viewed as a special case of our online nonconvex min-max problem, if $f_t(x, y)$ takes values independent of $y$. In such a degenerate case, our local NE-regret is equivalent to the local regret analyzed in Hazan et al. (2017a); Hallak et al. (2021). Consequently, the adversarial example that incurs the local regret of $\Omega(\frac{T}{w^2})$ constructed in Hallak et al. (2021) can also serve as a worst case example for our online noncovex min-max setting. Moreover, under the same assumption made in Hazan et al. (2017a) (which is more restrictive than our assumption here), we achieve a robust regret upper bound of $O(\frac{T}{w^2})$ (as discussed in the previous paragraph), which matches the worst-case lower bound, indicating that our bound Theorem 1 for online nonconvex min-max problem is optimal.

**Comparison to offline min-max optimization.** When the environment is fixed, i.e. $f_t \equiv f$ or $T = 1$ with $w = 1$, our problem specializes to offline min-max optimization and $V_w^1[T] = V_w^2[T] = 0$ will disappear from our results. Therefore, an immediate implication from our theorems is that GDA is guaranteed to find $\epsilon$-stationary point with $O(\kappa^3\epsilon^{-2})$ iteration complexity. The best known complexity bound for GDA in offline min-max optimization is $O(\kappa^2\epsilon^{-2})$ (Lin et al., 2020a). However, as we discussed in Section 4.1, TSODA aims to output $\mathbf{x}$ with last-iterate type guarantee, which is a stronger notion than that considered in Lin et al. (2020a), where GDA are only guaranteed to visit an $\epsilon$-stationary point within a certain number of iterations. Thus, these results are not directly comparable.

## 5 TSODA with Stochastic First-order Oracle

In this section, we extend our online min-max framework to an online stochastic version. This setting is motivated by the fact that, in real world application, such as training a neural network, an oracle with access to the gradient of loss function is hard to reach. Instead, a stochastic first-order oracle (SFO) is used to approximate the ground truth gradient. Similar settings have been studied in Nemirovski et al. (2009); Hazan et al. (2017a); Hallak et al. (2021). Specifically, the formal SFO definition is as follows.

**Definition 3** (Stochastic first-order oracle). *A stochastic first-order oracle (SFO) is a function $\mathcal{S}_\sigma$ such that, given a point $(\mathbf{x}, \mathbf{y}) \in \mathbb{R}^m \times \mathcal{Y}$, a random seed $\zeta$, and a smooth function $h : \mathbb{R}^m \times \mathcal{Y} \to \mathbb{R}$ satisfies:*

- $\mathcal{S}_\sigma(\mathbf{x}, \mathbf{y}; \zeta, h)$ *is an unbiased estimate of* $\nabla h(\mathbf{x}, \mathbf{y})$ : $\mathbb{E}(\mathcal{S}(\mathbf{x}, \mathbf{y}; \zeta, h) - \nabla h(\mathbf{x}, \mathbf{y})) = 0$;
- $\mathcal{S}_\sigma(\mathbf{x}, \mathbf{y}; \zeta, h)$ *has variance bounded by* $\sigma^2 > 0$ : $\mathbb{E}\left(\|\mathcal{S}(\mathbf{x}, \mathbf{y}; \zeta, h) - \nabla h(\mathbf{x}, \mathbf{y})\|^2\right) \le \sigma^2$.

### 5.1 Proposed Algorithm

With the above definition of SFO, we introduce the stochastic version of Algorithm 1, named TSODA-SFO (see Algorithm 2). Similarly, TSODA-SFO also follows the-leader iterates using two-time scale GDA. Taking the noise brought by SFO into consideration, nested loops and special stopping criterion (Stop Condition 2) are modified accordingly. Specially, (*i*) SFO results in different coefficients in stop criterion compared to TSODA. (*ii*) The stopping criterion in TSODA-SFO only ensures that $\|\nabla\Phi_{t,w}(\mathbf{x}_{t+1})\|^2$ is bounded by the threshold plus the variation of SFO. But the variation here does not play an important role, since sliding windows serve variance reduction purpose to reduce the variation in the final expected regret.

*Stop Condition* 2. The terminating condition for Algorithm 2 is:

$$2\left((\frac{2\kappa}{\eta_\mathbf{y}} + \ell)(1 + \ell\eta_\mathbf{y})\right)^2 \|\mathbf{y}_t^k - \mathcal{P}_\mathcal{Y}\left(\mathbf{y}_t^k + \eta_\mathbf{y}G_{\mathbf{y},t}^k\right)\|^2 + \|G_{\mathbf{x},t}^k\|^2 \le \delta^2/3w^2. \tag{8}$$

---

**Algorithm 2** TSODA with Stochastic First-order Oracle (TSODA-SFO)

---

**Input:** window size $w \geq 1$, stepsizes $(\eta_{\mathbf{x}}, \eta_{\mathbf{y}})$, tolerance $\delta > 0$
**Initialization:** $(\mathbf{x}_1, \mathbf{y}_1)$

1: **for** $t = 1$ to $T$ **do**
2:     Cost function $f_t : \mathbb{R}^m \times \mathbb{R}^n \to \mathbb{R}$ is updated;
3:     Sample $\tilde{\nabla} f_t (\mathbf{x}_t, \mathbf{y}_t) \leftarrow \mathcal{S}_{\sigma/w} (\mathbf{x}_t, \mathbf{y}_t; \zeta, f_t)$
4:     Set $\tilde{\nabla} F_{t,w} (\mathbf{x}_t, \mathbf{y}_t) = \tilde{\nabla} F_{t-1,w} (\mathbf{x}_t, \mathbf{y}_t) + \frac{1}{w} (\tilde{\nabla} f_{t-w} (\mathbf{x}_t, \mathbf{y}_t) - \tilde{\nabla} f_t (\mathbf{x}_t, \mathbf{y}_t))$
5:     Set $\mathbf{x}_t^0 = \mathbf{x}_t$, $\mathbf{y}_t^0 = \mathbf{y}_t$, $G_{\mathbf{x},t}^0 = \tilde{\nabla}_{\mathbf{x}} F_{t,w} (\mathbf{x}_t, \mathbf{y}_t)$, $G_{\mathbf{y},t}^0 = \tilde{\nabla}_{\mathbf{y}} F_{t,w} (\mathbf{x}_t, \mathbf{y}_t)$, $k = 0$
6:     **while** Equation (8) in Stop Condition 2 is not satisfied **do**
7:         $\mathbf{x}_t^{k+1} \leftarrow \mathbf{x}_t^k - \eta_{\mathbf{x}} G_{\mathbf{x},t}^k$
8:         $\mathbf{y}_t^{k+1} \leftarrow \mathcal{P}_{\mathcal{Y}} (\mathbf{y}_t^k + \eta_{\mathbf{y}} G_{\mathbf{y},t}^k)$
9:         Sample $\tilde{\nabla} f_i(\mathbf{x}_t^{k+1}, \mathbf{y}_t^{k+1}) \leftarrow \mathcal{S}_{\frac{\sigma}{w}} (\mathbf{x}_t^{k+1}, \mathbf{y}_t^{k+1}; \zeta, f_i)$ for $i = t - w + 1, \cdots, t$;
10:         Set $G_t^{k+1} := (G_{t,\mathbf{x}}^{k+1}, G_{t,\mathbf{y}}^{k+1}) = \frac{1}{w} \sum_{i=t-w+1}^t \tilde{\nabla} f_i(\mathbf{x}_t^{k+1}, \mathbf{y}_t^{k+1})$
11:         $k \leftarrow k + 1$
12:     **end while**
13:     $\mathbf{x}_{t+1} = \mathbf{x}_t^k$, $\mathbf{y}_{t+1} = \mathbf{y}_t^k$, and $\tilde{\nabla} F_{t,w}(\mathbf{x}_{t+1}, \mathbf{y}_{t+1}) = G_t^k$
14: **end for**

---

## 5.2 Theoretical Guarantees

Denote $\tau_t$ as the number of iterations of inner-loop at round $t$ and thus $\tau = \sum_{t=1}^T \tau_t$. Given the SFO and the inner-loop termination condition in eq. (8), one immediate question is whether Algorithm 2 terminates in finite time. To this end, we first establish that for each round $t$, the inner-loop terminates with finite iterations $\tau_t$ provided that $\delta$ is not too small (recall that $\delta$ is the tolerance for stopping criterion).

**Theorem 3** (Finite Iteration with SFO). *Let $\kappa = \ell/\mu$ denote the condition number, and let the stepsizes be chosen as $\eta_{\mathbf{x}} = \Theta\left(1/\kappa^3 \ell\right)$ and $\eta_{\mathbf{y}} = \Theta(1/\ell)$. Under Assumptions 1-3, for any $t \in [T]$, if $\delta, w$ and $\sigma$ satisfy that $\delta^2 = O(\frac{\kappa^4 \ell^2 \sigma^2}{w})$, then $\tau_t$ and $\tau$ are finite with high probability. Specially, when $K \in \mathbb{R}$ is large enough, $\mathbb{P}(\tau_t > K) = O(1/K)$.*

With the finite step stopping guarantee on hand, we next characterize the performance of TSODA-SFO with expected local NE-regret formally in terms of $w, T, V_w^1[T], V_w^2[T]$.

**Theorem 4** (Expected Local NE-Regret with SFO). *Under the setting of Theorem 3, TSODA-SFO enjoys the following expected local NE regret bound:*

$$\mathbb{E} \left[ \Re_w^{NE}(T) \right] \leq \frac{T}{w^2} \left( 3\delta^2 + \frac{(360\kappa^2 + 9)\sigma^2}{w} \right) + \frac{3\kappa^2}{(w-1)^2} V_w^2[T] + \frac{3}{w^2} V_w^1[T].$$

Beyond the previous regret analysis, we next provide an upper bound on the overall iteration complexity of SODA-SFO. Similar to Li & Orabona (2019); Hallak et al. (2021), we adopt the following stronger boundness assumption on the SFO to control the noise caused by SFO calls in the stochastic settings.

**Assumption 4.** *Given any point $(\mathbf{x}, \mathbf{y}) \in \mathbb{R}^m \times \mathcal{Y}$, random seed $\zeta$, and smooth function $h$: $\mathbb{R}^m \times \mathcal{Y}$, the SFO defined in Definition 3 satisfys that $\|\mathcal{S}(\mathbf{x}, \mathbf{y}; \zeta, h) - \nabla h(\mathbf{x}, \mathbf{y})\|^2 \leq \sigma^2$.*

*Remark* 3. We remark here that Theorems 3 and 4 do not require Assumption 4, and Theorem 3 provide the finite iteration guarantee with high probability and Theorem 4 provides an upper bound for expected regret. With Assumption 4, which is slightly stronger than the assumptions in Definition 3, we are able to provide the following deterministic bound on iterations and the number of SFO calls as in Theorem 5. Furthermore, Theorem 5 can provide deterministic guarantees rather than high probability guarantees because Assumption 4 controls the variation of noise in an absolute and deterministic manner.

**Theorem 5** (Iterations and SFO calls bounds). *Under the setting of Theorem 3 and Assumption 4, and suppose that $\delta^2 > 8064\kappa^4\sigma^2$. Then the total number of iterations satisfies*

$$\tau \leq \frac{1}{\eta_{\mathbf{x}}} \frac{2MTw + \frac{9\delta^2 T}{\ell} + \frac{72\ell w^2}{\mu^2(w-1)^2}V_w^2[T] + w^2 M + \frac{5\ell D^2 w^2}{32}}{\left(\frac{\delta^2}{3} - 2688\kappa^4\sigma^2\right)}$$

*Furthermore, the number of SFO calls is bounded by $O(w\tau)$.*

The above results also provide a robust guarantee for TSODA-SFO, where TSODA-SFO achieves an expected regret of $O(\frac{T}{w^2})$ with at most $O(Tw)$ iterations and hence $O(Tw^2)$ calls of SFO, as long as $V_w^1[T]$ and $V_w^2[T]$ scale with $O(T)$. Following the similar discussions from Remark 2 and Section 4.2, such condition can hold with relaxed assumptions depending on nonstationarity.

Specially, if the variance of SFO defined in Definition 3 is zero, then SFO reduces to perfect first order feedback. Hence, as discussed in Section 4.2, the adversarial example provided by Hazan et al. (2017a) is also applicable to the stochastic setting, and thus indicates that the expected regret $O(\frac{T}{w^2})$ reaches optimality. If the set $\mathcal{Y}$ is a singleton, online nonconvex min-max problem with SFO reduces to the online nonconvex problem with SFO. In this case, the term consisting of $V_w^2[T]$ will disappear in our analysis, and our theorems recover the results in Hallak et al. (2021).

## 6 Proof Overview

In this section, we will outline the regret and iteration analyses for TSODA (Theorems 1 and 2).

### 6.1 Key Ideas in the Proof of Theorem 1

We can directly decompose the local NE-regret $\Re_w^{NE}$ by Cauchy-Schwarz inequality as follows:

$$\Re_w^{NE} \leq \underbrace{3\sum_{t=1}^{T}\|\nabla\Phi_{t-1,w}(\mathbf{x}_t)\|^2}_{\text{Optimization error}} + \underbrace{3\sum_{t=1}^{T}\|\nabla_{\mathbf{x}}F_{t-1,w}(\mathbf{x}_t, \mathbf{y}_{t,w}^*(\mathbf{x}_t)) - \nabla_{\mathbf{x}}F_{t-1,w}(\mathbf{x}_t, \mathbf{y}_{t-1,w}^*(\mathbf{x}_t))\|^2}_{\text{Variability of }\mathbf{y}}$$

$$+ \underbrace{\frac{3}{w^2}\sum_{t=1}^{T}\|\nabla_{\mathbf{x}}f_t(\mathbf{x}_t, \mathbf{y}_{t,w}^*(\mathbf{x}_t)) - \nabla_{\mathbf{x}}f_{t-w}(\mathbf{x}_t, \mathbf{y}_{t,w}^*(\mathbf{x}_t))\|^2}_{\text{Variability of }\mathbf{x}}$$

In the following part, we will interpret each error term and provide a high-level overview of how to control them.

*Optimization error.* This term arises due to the overarching strategy of TSODA, which is to perform two-timescale GDA at each round $t$ in order to seek an approximate stationary point of $\Phi_{t,w}(\cdot)$. The following key lemma shows that if Stop Condition 1 is satisfied, then $\|\nabla\Phi_{t-1,w}(\mathbf{x}_t)\|^2 \leq \frac{\delta^2}{w^2}$ when TSODA enters the $t$-th round. This implies that the optimization error can be controlled by $\frac{3T\delta^2}{w^2}$.

**Lemma 6.1.** *Given a pair $(\mathbf{x}, \mathbf{y}) \in \mathbb{R}^m \times \mathcal{Y}$, for $t \in [T]$ and $w > 0$, it holds that*

$$\|\nabla\Phi_{t,w}(\mathbf{x})\|^2 \leq 2\left((\frac{2\kappa}{\eta_{\mathbf{y}}} + \ell)(1 + \ell\eta_{\mathbf{y}})\right)^2 \|\mathbf{y} - \mathcal{P}_{\mathcal{Y}}(\mathbf{y} + \eta_{\mathbf{y}}\nabla_{\mathbf{y}}F_{t,w}(\mathbf{x}, \mathbf{y}))\|^2 + 2\|\nabla_{\mathbf{x}}F_{t,w}(\mathbf{x}, \mathbf{y})\|^2.$$

*Variability of $\mathbf{x}$ and $\mathbf{y}$.* These errors occur since the model $\{f_{t-i}\}_{i=0}^{w-1}$ used for evaluating $(\mathbf{x}_t, \mathbf{y}_t)$ is different from the training model $\{f_{t-i}\}_{i=1}^{w}$, which corresponds to the level of the variation in the environment. i). The variability of $\mathbf{x}$ can be directly bounded by the sliding-window variation $V_w^1[T]$ by definition. ii). We show in Lemma B.1 that $F_{t,w}(\mathbf{x}, \mathbf{y})$ is $\ell$-smooth and $\mathbf{y}_{t,w}^*(\cdot)$ is $\kappa$-Lipschitz, and hence the variability of $\mathbf{y}$ can be further bounded by $\frac{3\kappa^2}{(w-1)^2}\|\nabla_{\mathbf{y}}f_t(\mathbf{x}_t, \mathbf{y}_{t,w}^*(\mathbf{x}_t)) - \nabla_{\mathbf{y}}f_{t-w}(\mathbf{x}_t, \mathbf{y}_{t-1,w}^*(\mathbf{x}_t))\|^2$, which is controlled by $V_w^2[T]$.

## 6.2 Key Ideas of Theorem 2

The proof of Theorem 2 can be divided into two parts: the inner-loop and outer-loop analysis.

**Inner-loop Analysis.** Denote the sequence generated in the inner-loop at time $t \in [T-1]$ by

$$\mathbf{x}_t^0 = \mathbf{x}_t, \quad \mathbf{x}_t^{k+1} \leftarrow \mathbf{x}_t^k - \eta_{\mathbf{x}} \nabla_{\mathbf{x}} F_{t,w}\left(\mathbf{x}_t^k, \mathbf{y}_t^k\right);$$
$$\mathbf{y}_t^0 = \mathbf{y}_t, \quad \mathbf{y}_t^{k+1} \leftarrow \mathcal{P}_{\mathcal{Y}}\left(\mathbf{y}_t^k + \eta_{\mathbf{y}} \nabla_{\mathbf{y}} F_{t,w}\left(\mathbf{x}_t^k, \mathbf{y}_t^k\right)\right).$$

Let $\tau_t$ be the number of times the gradient update is executed at the $t$-th iteration. For convenience, denote $\tau_T = 0$. Note that $\mathbf{x}_t^{\tau_t} = \mathbf{x}_{t+1}$ and $\mathbf{y}_t^{\tau_t} = \mathbf{y}_{t+1}$. In such a two time-scale setting, i.e. $\eta_{\mathbf{x}} \neq \eta_{\mathbf{y}}$, the movement of $\mathbf{x}_t^k$ is slower than $\mathbf{y}_t^k$, and the $\kappa$-Lipschitzness of $\mathbf{y}_{t,w}^{\star}(\cdot)$ indicates that $\mathbf{y}_{t,w}^{\star}(\mathbf{x}_t^k)$ also moves slowly. The inner-loop of TSODA can be viewed as conducting gradient ascent on a strongly-concave function $F_{t,w}(\mathbf{x}_t^k, \cdot)$, which changes slowly. Following the standard analysis of offline nonconvex min-max optimization Lin et al. (2020b), we can establish the following descent property:

$$\frac{\eta_{\mathbf{x}}}{8} \sum_{j=0}^{\tau_t-1} \left[\left\|\nabla_{\mathbf{x}} F_{t,w}\left(\mathbf{x}_t^j, \mathbf{y}_t^j\right)\right\|^2 + (6\kappa\ell)^2 \|\mathbf{y}_t^{j+1} - \mathbf{y}_t^j\|^2\right] \leq (\Phi_{t,w}\left(\mathbf{x}_t\right) - \Phi_{t,w}\left(\mathbf{x}_{t+1}\right)) + \frac{9\ell}{2}\delta_{t,w}^0, \tag{9}$$

where $\delta_{t,w}^0 = \left\|\mathbf{y}_{t,w}^{\star}\left(\mathbf{x}_t^0\right) - \mathbf{y}_t^0\right\|^2$ measures the distance between $\mathbf{y}_t^0$ and the optimal solution for $y$-player given $\mathbf{x}_t^0$ at the beginning of round $t$. Notice that the LHS of eq. (9) is the quantity in Stop Condition 1 when $\eta_{\mathbf{y}} = \frac{1}{\ell}$. Since TSODA performs GDA at round $t$ only when Stop Condition 1 does not meet, we can further lower-bound the LHS of eq. (9) by $\frac{\eta_{\mathbf{x}}}{8} \cdot \frac{\delta^2}{2w^2} \cdot \tau_t$, and obtain

$$\frac{\eta_{\mathbf{x}} \delta^2 \tau_t}{16w^2} \leq (\Phi_{t,w}\left(\mathbf{x}_t\right) - \Phi_{t,w}\left(\mathbf{x}_{t+1}\right)) + \frac{9\ell}{2}\delta_{t,w}^0. \tag{10}$$

**Outer-loop Analysis.** By decomposing $\Phi_{T,w}(\mathbf{x}_T) = \sum_{t=1}^{T}(\Phi_{t,w}(\mathbf{x}_t) - \Phi_{t-1,w}(\mathbf{x}_{t-1}))$ and rearranging terms, we obtain:

$$\sum_{t=1}^{T-1} \Phi_{t,w}(\mathbf{x}_t) - \Phi_{t,w}(\mathbf{x}_{t+1}) \leq \frac{1}{w} \sum_{t=1}^{T} \left(f_t(\mathbf{x}_t, \mathbf{y}_{t,w}^*(\mathbf{x}_t)) - f_{t-w}(\mathbf{x}_t, \mathbf{y}_{t,w}^*(\mathbf{x}_t))\right) - \Phi_{T,w}(\mathbf{x}_T). \tag{11}$$

Substituting eq. (10) over $t \in [T]$ into the above inequality, we obtain

$$\frac{\eta_{\mathbf{x}} \delta^2 \tau}{16w^2} \leq \underbrace{\frac{1}{w} \sum_{t=1}^{T} \left(f_t(\mathbf{x}_t, \mathbf{y}_{t,w}^*(\mathbf{x}_t)) - f_{t-w}(\mathbf{x}_t, \mathbf{y}_{t,w}^*(\mathbf{x}_t))\right) - \Phi_{T,w}(\mathbf{x}_T)}_{V_1} + \underbrace{\frac{9\ell}{2} \sum_{t=1}^{T} \delta_{t,w}^0}_{V_2}.$$

The $V_1$ term can be bounded by considering the boundness of $f_t$. As for the $V_2$ term, we can upper bound $\delta_{t,w}^0$ for $t > 1$ using $\|\mathbf{y}_{t,w}^*(x_t) - \mathbf{y}_{t-1,w}^*(x_t)\|^2 + \|\mathbf{y}_{t-1,w}^*(x_t) - \mathbf{y}_t^0\|^2$. This quantity can be further controlled by the sliding-window variation in $\mathbf{y}$ and the tolerance $\delta$ in Stop Condition 1 separately. Note that $\delta_{1,w}^0$ can be directly bounded by $D$, which is the diameter of $\mathcal{Y}$.

**Significance of Techniques.** Based on the sketch of the analysis provided in this section, we can now delve into the technical differences between our work and the general online nonconvex games, specifically the work presented in Hazan et al. (2017a). One key difference is that we lack direct access to the first-order oracle of $\Phi_{t,w}$, whereas such information is available in Hazan et al. (2017a). Consequently, while our outer-loop analysis draws inspiration from Hazan et al. (2017a), we must address the challenge of limited knowledge of the first-order oracle and develop novel stop conditions tailored to the min-max setting. More importantly, our inner loop features a more intricate min-max structure, which requires further technical development to handle the dynamics of two players. The analysis presented in Hazan et al. (2017a), which focuses on a single player, cannot be directly extended to our setting. As a result, specialized techniques for min-max optimization must be employed to analyze the iteration complexity in the inner loop.

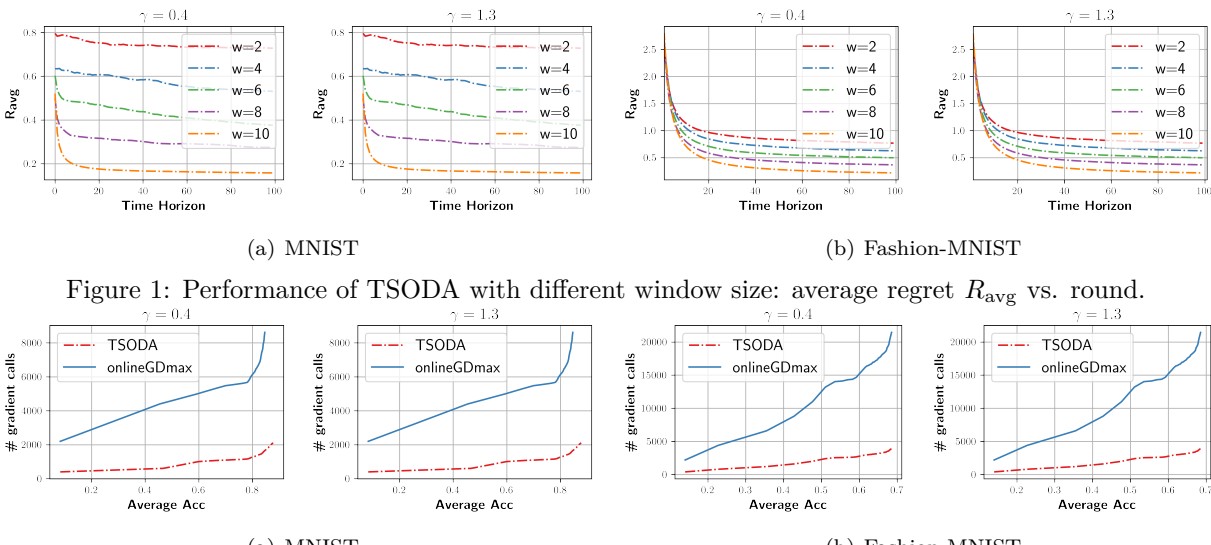

(a) MNIST

(b) Fashion-MNIST

Figure 1: Performance of TSODA with different window size: average regret $R_{\text{avg}}$ vs. round.

(a) MNIST

(b) Fashion-MNIST

Figure 2: Comparison of TSODA and onlineGDmax: number of gradient calls vs. average accuracy.

# 7 Experiments

In this section, we evaluate the efficiency of the proposed TSODA algorithm and verify the theoretical results through numerical simulations.

We consider the min-max problem of training an empirical Wasserstein robustness model (WRM) (Sinha et al., 2017), which has the following form[2]:

$$\min_{\mathbf{x}} \max_{\{\mathbf{y}_i\}_{i=1}^N} \mathcal{L}(\mathbf{x}, \mathbf{y}; \mathcal{D}) \triangleq \frac{1}{N} \sum_{(\xi_i, z_i) \in \mathcal{D}} \left[ \ell\left(h_{\mathbf{x}}(\mathbf{y}_i), z_i\right) - \gamma \left\|\xi_i - \mathbf{y}_i\right\|^2 \right], \tag{12}$$

where $\ell$ is the cross-entropy loss function, $N$ is the number of training samples, $\mathbf{x}$ is the network parameter, $(\xi_i, z_i) \in \mathcal{D}$ corresponds to the $i$-th data sample and label, respectively, and $\mathbf{y}_i$ is the adversarial sample corresponding to $\xi_i$. Denote $\{\mathbf{y}_i\}_{i=1}^N$ as $\mathbf{y}$.

**Training Settings.** The real-world datasets we consider are MNIST (Deng, 2012) and Fashion-MNIST (Xiao et al., 2017), each containing 60k samples. We simulate the online WRM model as follows. We randomly split the given dataset into $T$ pieces $\{\mathcal{D}_t\}_{t=1}^T$, and the learner sequentially receives $\mathcal{D}_t$. At each round $t$, $f_t(\mathbf{x}, \mathbf{y}) = \mathcal{L}(\mathbf{x}, \mathbf{y}; \mathcal{D}_t)$. We choose $T = 100$ for the online setting. The network architecture mainly follows Sinha et al. (2017), which consists of three convolution blocks with filters of size $8 \times 8, 6 \times 6$ and $5 \times 5$ respectively activated by ELU function (Clevert et al., 2015), then followed by a fully connected layer and softmax output. Furthermore, we set the adversarial perturbation $\gamma \in \{0.4, 1.3\}$, which is consistent with Sinha et al. (2017).

**Metrics.** Since we do not have access to the first-order oracle of $\nabla \Phi_{t,w}$ in practice, two alternative performance metrics are considered, which capture the essence of the online setting and are consistent with the definition of our local NE-regret. The first metric is the stronger notion we utilize in the stop criterion, which provides an upper bound for $\|\nabla \Phi_{t,w}(\mathbf{x}_t)\|^2$. Observing that the projected gradient of $\mathbf{y}$ does not change significantly in experiments, we only compute $\|\nabla_{\mathbf{x}} F_{t,w}(\mathbf{x}_t, \mathbf{y}_t)\|^2$ and report the average $R_{\text{avg}} \triangleq \frac{1}{t} \sum_{j=1}^t \|\nabla_{\mathbf{x}} F_{j,w}(\mathbf{x}_j, \mathbf{y}_j)\|^2$ of these at each round $t$, which serves as an approximation of $\frac{1}{t} \Re_w^{NE}(t)$. The second metric is the average accuracy, where we evaluate the test accuracy of output $(\mathbf{x}_t, \mathbf{y}_t)$ from the last round on the newly coming $\mathcal{D}_t$ and report the average from round 1 to $t$.

**The Effect of Window Size** $w$. In Figure 1, we plot $R_{\text{avg}}$ of TSODA on MNIST and Fashion-MNIST with different $w$. It can be observed that as $w$ increases from 2 to 10, the local regret becomes smaller, which verifies the bound in Theorem 1 and justifies the usage of large window size.

---

[2]Note that we can choose sufficiently large $\gamma > 0$ to make the maximization part be strongly-concave.

**TSODA vs. Baseline Algorithm.** To further investigate the performance of TSODA, we conduct experiments to compare it with a baseline algorithm. Note that to our best knowledge, there has been no existing formal studies on the performance of any developed algorithm for online nonconvex min-max problems. Here, we consider a baseline algorithm, which is a natural extension of the well-known offline min-max method GDmax (Jin et al., 2020) to the online framework, named *onlineGDmax*. Specifically, onlineGDmax replaces the inner-loop procedure of TSODA by the nested-loop GDmax, i.e., at each iteration in the inner-loop of round $t$, onlineGDmax will firstly maximize the function by multi-step gradient ascent for $\mathbf{y}$, which is 10 steps in our setting, and then perform one-step GD for $\mathbf{x}$. Typically, the stepsizes for GDmax are chosen to be equal, i.e. $\eta_{\mathbf{x}} = \eta_{\mathbf{y}}$ (Sinha et al., 2017). In Figure 2, TSODA achieves similar accuracy to onlineGDmax but with significantly fewer gradient calls, which demonstrates the efficiency of our approach.

## 8 Conclusions

This paper provides the first analysis for the online nonconvex-concave min-max optimization problem. We introduced a novel notion of local Nash Equilibrium regret to capture the nonconvexity and nonstationary of the environment. We developed and analyzed algorithms TSODA and its stochastic version with respect to the proposed notions of regret, establishing favorable regret and complexity guarantees. Furthermore, we conduct experiments with real-world data to validate the theoretical statements and show its superiority in practice.

## Acknowledgments

The work of Yu Huang and Longbo Huang is supported by the Technology and Innovation Major Project of the Ministry of Science and Technology of China under Grant 2020AAA0108400 and 2020AAA0108403, and Tsinghua Precision Medicine Foundation 10001020109.

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

# Appendices

## A   Concrete Toy Example for Section 3.1

Consider a two-player game characterized by time-varying loss functions $\{f_t\}_{t=1}^T$. For any $t \in [T]$, we define the function as:

$$f_t(\mathbf{x}, \mathbf{y}) = a_t \mathbf{x}^2 - b_t \mathbf{y}^2 \quad \mathbf{x} \in \mathbb{R}, \mathbf{y} \in [-1, 1] \tag{13}$$

Here, both $a_t$ and $b_t$ are strictly positive. It is evident that $f_t$ is strongly-convex with respect to $\mathbf{x}$ and strongly-concave with respect to $\mathbf{y}$. We can straightforwardly compute the min-max value of the problem $\min_{\mathbf{x} \in \mathbb{R}} \max_{\mathbf{y} \in [-1,1]} f_t(\mathbf{x}, \mathbf{y})$ to be 0, which corresponds to the optimal solution $(\mathbf{x}_t, \mathbf{y}_t) = (0, 0)$.

Referencing Zhang et al. (2022), the dynamic Nash Equilibrium (NE)-regret is defined as:

$$\Re^{\text{dyn}}(T) := | \sum_{t=1}^{T} f_t(\mathbf{x}_t, \mathbf{y}_t) - \sum_{t=1}^{T} \min_{\mathbf{x} \in \mathbb{R}} \max_{\mathbf{y} \in [-1,1]} f_t(\mathbf{x}, \mathbf{y})|.$$

We further simplify by setting $\mathbf{y}_t = \arg\max_{\mathbf{y} \in [-1,1]} f_t(\mathbf{x}, \mathbf{y})$, as the maximization problem can be efficiently solved even in our nonconvex-strongly-concave setting. Then, for $f_t$ as defined in eq. (13), we obtain $\Re^{\text{dyn}}(T) = \sum_{t=1}^{T} a_t x_t^2$.

On the other hand, when $w = 1$, $\nabla \Phi_{t,w}(\mathbf{x}_t) = 2a_t \mathbf{x}_t$, then the local NE-regret is

$$\mathfrak{R}_1^{\text{NE}}(T) = \sum_{t=1}^{T} \|\nabla \Phi_{t,1}(\mathbf{x}_t)\|^2 = 2 \sum_{t=1}^{T} a_t^2 \mathbf{x}_t^2.$$

Combining these results, we obtain:

$$\Re^{\text{dyn}}(T) = \sum_{t=1}^{T} \frac{a_t^2}{a_t} x_t^2 \leq \frac{1}{\min_{t \in [T]} a_t} \sum_{t=1}^{T} a_t^2 \mathbf{x}_t^2 \leq C \mathfrak{R}_1^{\text{NE}}(T)$$

where $C = \frac{1}{2 \min_{t \in [T]} a_t}$ is some positive constant.

From the above derivations, it is clear that in this specific toy case, a lower local NE-regret implies a lower dynamic NE-regret.

## B Missing Proof of Section 4

### B.1 Technical Lemma

Recall that $\Phi_{t,w}(\mathbf{x}) = \max_{\mathbf{y} \in \mathcal{Y}} F_{t,w}(\mathbf{x}, \mathbf{y})$ and $\mathbf{y}_{t,w}^*(\mathbf{x}) = \arg\max_{\mathbf{y} \in \mathcal{Y}} F_{t,w}(\mathbf{x}, \mathbf{y})$. In this section, we first present some technical lemmas to characterize the structure of the function $\Phi_{t,w}$ and $\mathbf{y}_{t,w}^*$ in the nonconvex-strongly-concave setting, which will be essential throughout the analysis.

**Lemma B.1.** $\Phi_{t,w}(\cdot)$ is $(\ell + \kappa\ell)$-smooth with $\nabla \Phi_{t,w}(\cdot) = \nabla_{\mathbf{x}} F_{t,w}(\cdot, \mathbf{y}_{t,w}^\star(\cdot))$. Also, $\mathbf{y}_{t,w}^\star(\cdot)$ is $\kappa$-Lipschitz.

*Proof.* Since the averaging maintains strongly-nonconcavity and smoothness, i.e. $F_{t,w}(\mathbf{x}, \mathbf{y})$ is still $\mu$-strongly-concave in $\mathbf{y}$ and $\ell$-smooth. Thus, the proof directly follow Lemma 4.3 in Lin et al. (2020a) and we omit the details. □

Furthermore, we derive the following lemma to provide the smoothness property of $\mathbf{y}_{t,w}^*(\mathbf{x})$ with respect to $t$. In other words, given any fixed $\mathbf{x}$, the movement of $\mathbf{y}_{t,w}^*(\mathbf{x})$ when $t$ changes can be controlled by the variability of environment of the sliding window.

**Lemma B.2.** For any $\mathbf{x} \in \mathbb{R}^m$, $t \in [T]$, it holds that

$$\left\| \mathbf{y}_{t-1,w}^*(\mathbf{x}) - \mathbf{y}_{t,w}^*(\mathbf{x}) \right\| \leq \frac{\left\| \nabla_{\mathbf{y}} f_{t,w}(\mathbf{x}, \mathbf{y}_{t,w}^*(\mathbf{x})) - \nabla_{\mathbf{y}} f_{t-w}(\mathbf{x}, \mathbf{y}_{t-1,w}^*(\mathbf{x})) \right\|}{\mu(w-1)}.$$

*Proof.* By the optimality of $\mathbf{y}_{t,w}^*(\mathbf{x})$ and $\mathbf{y}_{t-1,w}^*(\mathbf{x})$, for $\forall \mathbf{x}$, we have

$$(\mathbf{y} - \mathbf{y}_{t,w}^*(\mathbf{x}))^\top \nabla_{\mathbf{y}} F_{t,w}(\mathbf{x}, \mathbf{y}_{t,w}^*(\mathbf{x})) \leq 0, \forall \mathbf{y} \in \mathcal{Y}, \tag{14}$$

$$(\mathbf{y} - \mathbf{y}_{t-1,w}^*(\mathbf{x}))^\top \nabla_{\mathbf{y}} F_{t-1,w}(\mathbf{x}, \mathbf{y}_{t-1,w}^*(\mathbf{x})) \leq 0, \forall \mathbf{y} \in \mathcal{Y}. \tag{15}$$

Summing up Equation (14) with $\mathbf{y} = \mathbf{y}_{t-1,w}^*(\mathbf{x})$ and Equation (15) with $\mathbf{y} = \mathbf{y}_{t,w}^*(\mathbf{x})$ yields that

$$(\mathbf{y}_{t-1,w}^*(\mathbf{x}) - \mathbf{y}_{t,w}^*(\mathbf{x}))^\top (\nabla_{\mathbf{y}} F_{t,w}(\mathbf{x}, \mathbf{y}_{t,w}^*(\mathbf{x})) - \nabla_{\mathbf{y}} F_{t-1,w}(\mathbf{x}, \mathbf{y}_{t-1,w}^*(\mathbf{x}))) \leq 0. \tag{16}$$

By the definition of $F_{t,w}(\mathbf{x}, \mathbf{y})$, we have

$$\begin{aligned}
&\nabla_{\mathbf{y}} F_{t,w}(\mathbf{x}, \mathbf{y}_{t,w}^*(\mathbf{x})) - \nabla_{\mathbf{y}} F_{t-1,w}(\mathbf{x}, \mathbf{y}_{t-1,w}^*(\mathbf{x})) \\
&= \frac{1}{w} \sum_{i=0}^{w-1} \nabla_{\mathbf{y}} f_{t-i}(\mathbf{x}, \mathbf{y}_{t,w}^*(\mathbf{x})) - \frac{1}{w} \sum_{i=0}^{w-1} \nabla_{\mathbf{y}} f_{t-i-1}(\mathbf{x}, \mathbf{y}_{t-1,w}^*(\mathbf{x})) \\
&= \frac{1}{w} \left\{ \nabla_{\mathbf{y}} f_{t,w}(\mathbf{x}, \mathbf{y}_{t,w}^*(\mathbf{x})) - \nabla_{\mathbf{y}} f_{t-w}(\mathbf{x}, \mathbf{y}_{t-1,w}^*(\mathbf{x})) \right\} \\
&\quad + \frac{1}{w} \sum_{i=1}^{w-1} \left\{ \nabla_{\mathbf{y}} f_{t-i}(\mathbf{x}, \mathbf{y}_{t,w}^*(\mathbf{x})) - \nabla_{\mathbf{y}} f_{t-i}(\mathbf{x}, \mathbf{y}_{t-1,w}^*(\mathbf{x})) \right\}.
\end{aligned} \tag{17}$$

Since for any $t$ and fixed $\mathbf{x}$, the $f_t(\mathbf{x}, \cdot)$ is $\mu$-strongly-concave, we have

$$\begin{aligned}
&(\mathbf{y}_{t-1,w}^*(\mathbf{x}) - \mathbf{y}_{t,w}^*(\mathbf{x}))^\top \left\{ \nabla_{\mathbf{y}} f_{t-i}(\mathbf{x}, \mathbf{y}_{t-1,w}^*(\mathbf{x})) - \nabla_{\mathbf{y}} f_{t-i}(\mathbf{x}, \mathbf{y}_{t,w}^*(\mathbf{x})) \right\} \\
&\quad + \mu \left\| (\mathbf{y}_{t-1,w}^*(\mathbf{x}) - \mathbf{y}_{t,w}^*(\mathbf{x})) \right\|^2 \leq 0.
\end{aligned} \tag{18}$$

Plug Equations (17) and (18) into Equation (16), then we have

$$\begin{aligned}
&(\mathbf{y}_{t-1,w}^*(\mathbf{x}) - \mathbf{y}_{t,w}^*(\mathbf{x}))^\top \frac{1}{w} \left\{ \nabla_{\mathbf{y}} f_{t,w}(\mathbf{x}, \mathbf{y}_{t,w}^*(\mathbf{x})) - \nabla_{\mathbf{y}} f_{t-w}(\mathbf{x}, \mathbf{y}_{t-1,w}^*(\mathbf{x})) \right\} \\
&\quad + \frac{w-1}{w} \mu \left\| (\mathbf{y}_{t-1,w}^*(\mathbf{x}) - \mathbf{y}_{t,w}^*(\mathbf{x})) \right\|^2 \leq 0.
\end{aligned}$$

As a result

$$\begin{aligned}
&\frac{w-1}{w} \mu \left\| \mathbf{y}_{t-1,w}^*(\mathbf{x}) - \mathbf{y}_{t,w}^*(\mathbf{x}) \right\|^2 \\
&\leq -(\mathbf{y}_{t-1,w}^*(\mathbf{x}) - \mathbf{y}_{t,w}^*(\mathbf{x}))^\top \frac{1}{w} \left\{ \nabla_{\mathbf{y}} f_{t,w}(\mathbf{x}, \mathbf{y}_{t,w}^*(\mathbf{x})) - \nabla_{\mathbf{y}} f_{t-w}(\mathbf{x}, \mathbf{y}_{t-1,w}^*(\mathbf{x})) \right\} \\
&\leq \frac{1}{w} \left\| \mathbf{y}_{t-1,w}^*(\mathbf{x}) - \mathbf{y}_{t,w}^*(\mathbf{x}) \right\| \left\| \nabla_{\mathbf{y}} f_{t,w}(\mathbf{x}, \mathbf{y}_{t,w}^*(\mathbf{x})) - \nabla_{\mathbf{y}} f_{t-w}(\mathbf{x}, \mathbf{y}_{t-1,w}^*(\mathbf{x})) \right\},
\end{aligned}$$

where the last inequality follows from Cauchy-Schwartz inequality.

Finally, by some algebra manipulation, we finish the proof as following

$$\left\| \mathbf{y}_{t-1,w}^*(\mathbf{x}) - \mathbf{y}_{t,w}^*(\mathbf{x}) \right\| \leq \frac{\left\| \nabla_{\mathbf{y}} f_{t,w}(\mathbf{x}, \mathbf{y}_{t,w}^*(\mathbf{x})) - \nabla_{\mathbf{y}} f_{t-w}(\mathbf{x}, \mathbf{y}_{t-1,w}^*(\mathbf{x})) \right\|}{\mu(w-1)}.$$

$\square$

The next lemma provides an upper bound for the gradient norm of $\nabla \Phi_{t,w}$ in term of $\nabla F_{t,w}$, which justifies our design of stop conditions.

**Lemma B.3.** *Given a pair* $(\mathbf{x}, \mathbf{y}) \in \mathbb{R}^m \times \mathcal{Y}$, *for* $t \in [T]$ *and* $w > 0$, *it holds that*

$$\begin{aligned}
\|\nabla \Phi_{t,w}(\mathbf{x})\|^2 \leq &2 \left( (\frac{2\kappa}{\eta_{\mathbf{y}}} + \ell)(1 + \ell\eta_{\mathbf{y}}) \right)^2 \|\mathbf{y} - \mathcal{P}_{\mathcal{Y}}(\mathbf{y} + \eta_{\mathbf{y}} \nabla_{\mathbf{y}} F_{t,w}(\mathbf{x}, \mathbf{y}))\|^2 \\
&+ 2\|\nabla_{\mathbf{x}} F_{t,w}(\mathbf{x}, \mathbf{y})\|^2.
\end{aligned}$$

*Proof.* By Cauchy-Schwartz inequality, we have

$$\begin{aligned}
\|\nabla \Phi_{t,w}(\mathbf{x})\|^2 \leq &2\|\nabla \Phi_{t,w}(\mathbf{x}) - \nabla_{\mathbf{x}} F_{t,w}(\mathbf{x}, \mathbf{y})\|^2 + 2\|\nabla_{\mathbf{x}} F_{t,w}(\mathbf{x}, \mathbf{y})\|^2 \\
\leq &2\ell^2 \|\mathbf{y}_{t,w}^*(\mathbf{x}) - \mathbf{y}\|^2 + 2\|\nabla_{\mathbf{x}} F_{t,w}(\mathbf{x}, \mathbf{y})\|^2
\end{aligned}$$

where the last inequality holds by combining Lemma B.1 and the fact that $F_{t,w}$ is $\ell$-smooth.

To proceed the analysis, we need an important result about global error condition for proximal gradient algorithms from Drusvyatskiy & Lewis (2018), which is presented in Lemma B.4 for completeness. Since $F_{t,w}(\mathbf{x}, \cdot)$ is $\mu$-strongly-concave over $\mathcal{Y}$, we have

$$F_{t,w}(\mathbf{x}, \mathbf{y}) \geq \Phi_{t,w}(x) + \frac{\mu}{2} \|\mathbf{y} - \mathbf{y}_{t,w}^*(x)\|^2.$$

which implies $F_{t,w}(\mathbf{x}, \cdot)$ satisfies the quadratic growth condition. Then applying Lemma B.4, we obtain the error bound condition for $F_{t,w}(\mathbf{x}, \cdot)$ also holds. Specifically, in our setting, $\alpha = \mu$, $\beta = \ell$, and $\mathcal{G}_t$ degenerates to the projected gradient mapping with $t = \eta_{\mathbf{y}}$. Therefore,

$$
\begin{aligned}
\|\mathbf{y}_{t,w}^*(\mathbf{x}) - \mathbf{y}\| &\leq (\frac{2}{\mu} + \eta_{\mathbf{y}})(1 + \ell\eta_{\mathbf{y}}) \cdot \frac{1}{\eta_{\mathbf{y}}} \|\mathbf{y} - \mathcal{P}_{\mathcal{Y}}\left(\mathbf{y} + \eta_{\mathbf{y}} \nabla_{\mathbf{y}} F_{t,w}\left(\mathbf{x}, \mathbf{y}\right)\right)\| \\
&= (\frac{2\kappa}{\ell\eta_{\mathbf{y}}} + 1)(1 + \ell\eta_{\mathbf{y}}) \cdot \|\mathbf{y} - \mathcal{P}_{\mathcal{Y}}\left(\mathbf{y} + \eta_{\mathbf{y}} \nabla_{\mathbf{y}} F_{t,w}\left(\mathbf{x}, \mathbf{y}\right)\right)\|.
\end{aligned}
$$

Thus, we complete the proof. $\qquad\square$

**Lemma B.4** (Restate of Corollary 3.6 in Drusvyatskiy & Lewis (2018)). *Consider a closed, convex function $g : \mathbf{R}^n \to \overline{\mathbf{R}}$ and a $C^1$-smooth convex function $f : \mathbf{R}^n \to \mathbf{R}$ with $\beta$-Lipschitz continuous gradient. Denote the proximal mapping:*

$$\mathrm{prox}_{t,g}(x) := \underset{y \in \mathbf{R}^n}{\mathrm{argmin}} \left\{ g(y) + \frac{1}{2t} \|y - x\|^2 \right\}$$

*and the prox-gradient mapping:*

$$\mathcal{G}_t(x) := t^{-1}\left(x - \mathrm{prox}_{t,g}(x - t\nabla f(x))\right).$$

*Suppose that the function $\varphi := f + g$ has a nonempty set $S$ of minimizers and consider the following conditions:*

- *(Quadratic growth)*

$$\varphi(x) \geq \varphi^\star + \frac{\alpha}{2} \cdot \mathrm{dist}^2(x; S) \quad \text{for all } x \in [\varphi \leq \varphi^* + \nu] \tag{19}$$

- *(Error bound condition)*

$$\mathrm{dist}(x, S) \leq \gamma \|\mathcal{G}_t(x)\| \quad \text{is valid for all} \quad x \in [\varphi \leq \varphi^* + \nu] \tag{20}$$

*Then property (19) implies property (20) with $\gamma = \left(2\alpha^{-1} + t\right)(1 + \beta t)$. Conversely, condition (20) implies condition (19) with any $\alpha \in \left(0, \gamma^{-1}\right)$.*

## B.2   Local Regret: Proof of Theorem 1

*Proof of Theorem 1.* Recall the definition of $\Phi_{t,w}$ and notice that

$$\Phi_{t,w}(\mathbf{x}) = \max_{\mathbf{y} \in \mathcal{Y}} \frac{1}{w} \sum_{i=t-w+1}^{t} f_i(\mathbf{x}, \mathbf{y}) = \max_{\mathbf{y} \in \mathcal{Y}} \left[ F_{t-1,w}(\mathbf{x}, \mathbf{y}) + \frac{1}{w}(f_t(\mathbf{x}, \mathbf{y}) - f_{t-w}(\mathbf{x}, \mathbf{y})) \right].$$

Then

$$\|\nabla\Phi_{t,w}\left(\mathbf{x}_t\right)\|^2 = \left\|\nabla_{\mathbf{x}}F_{t,w}(\mathbf{x}_t,\mathbf{y}^*_{t,w}(\mathbf{x}_t))\right\|^2$$

$$= \left\|\nabla_{\mathbf{x}}F_{t-1,w}(\mathbf{x}_t,\mathbf{y}^*_{t-1,w}(\mathbf{x}_t)) + \nabla_{\mathbf{x}}F_{t-1,w}(\mathbf{x}_t,\mathbf{y}^*_{t,w}(\mathbf{x}_t)) - \nabla_{\mathbf{x}}F_{t-1,w}(\mathbf{x}_t,\mathbf{y}^*_{t-1,w}(\mathbf{x}_t))\right.$$

$$\left. + \frac{1}{w}\left(\nabla_{\mathbf{x}}f_t\left(\mathbf{x}_t,\mathbf{y}^*_{t,w}(\mathbf{x}_t)\right) - \nabla_{\mathbf{x}}f_{t-w}\left(\mathbf{x}_t,\mathbf{y}^*_{t,w}(\mathbf{x}_t)\right)\right)\right\|^2$$

$$\leq 3\left\|\nabla\Phi_{t-1,w}\left(\mathbf{x}_t\right)\right\|^2 + \frac{3\kappa^2}{(w-1)^2}\|\nabla_{\mathbf{y}}f_t\left(\mathbf{x}_t,\mathbf{y}^*_{t,w}(\mathbf{x}_t)\right) - \nabla_{\mathbf{y}}f_{t-w}\left(\mathbf{x}_t,\mathbf{y}^*_{t-1,w}(\mathbf{x}_t)\right)\|^2$$

$$+ \frac{3}{w^2}\|\nabla_{\mathbf{x}}f_t\left(\mathbf{x}_t,\mathbf{y}^*_{t,w}(\mathbf{x}_t)\right) - \nabla_{\mathbf{x}}f_{t-w}\left(\mathbf{x}_t,\mathbf{y}^*_{t,w}(\mathbf{x}_t)\right)\|^2, \tag{21}$$

where the second term in last inequality follows from that

$$\|\nabla_{\mathbf{x}}F_{t-1,w}(\mathbf{x}_t,\mathbf{y}^*_{t,w}(\mathbf{x}_t)) - \nabla_{\mathbf{x}}F_{t-1,w}(\mathbf{x}_t,\mathbf{y}^*_{t-1,w}(\mathbf{x}_t))\|$$

$$\leq \ell\|\mathbf{y}^*_{t,w}(\mathbf{x}_t) - \mathbf{y}^*_{t-1,w}(\mathbf{x}_t)\|$$

$$\stackrel{(a)}{\leq} \frac{\kappa\left\|\nabla_{\mathbf{y}}f_{t,w}(\mathbf{x},\mathbf{y}^*_{t,w}(\mathbf{x})) - \nabla_{\mathbf{y}}f_{t-w}(\mathbf{x},\mathbf{y}^*_{t-1,w}(\mathbf{x}))\right\|}{(w-1)}$$

where $(a)$ is implied by Lemma B.2.

Moreover, for the first term in Equation (21), by Lemma B.3, and the stop condition, we obtain

$$\|\nabla\Phi_{t-1,w}\left(\mathbf{x}_t\right)\|^2 \leq \frac{\delta^2}{w^2}.$$

Summing over $t = 1,\cdots,T$, and combining the definition of variation measures $V^1_w[T]$ and $V^2_w[T]$, then we have

$$\Re^{NE}_w(T) = \sum_{t=1}^{T}\|\Phi_{t,w}(\mathbf{x}_t)\|^2 \leq \frac{3}{w^2}(T\delta^2 + \frac{(\kappa w)^2}{(w-1)^2}V^2_w[T] + V^1_w[T]).$$

$\square$

### B.3 Oracle Queries: Proof of Theorem 2

Denote the sequence generated in the inner-loop at time $t \in [T-1]$ by

$$\mathbf{x}^0_t = \mathbf{x}_t \quad \mathbf{x}^{k+1}_t \leftarrow \mathbf{x}^k_t - \eta_{\mathbf{x}}\nabla_{\mathbf{x}}F_{t,w}\left(\mathbf{x}^k_t,\mathbf{y}^k_t\right)$$

$$\mathbf{y}^0_t = \mathbf{y}_t \quad \mathbf{y}^{k+1}_t \leftarrow \mathcal{P}_{\mathcal{Y}}\left(\mathbf{y}^k_t + \eta_{\mathbf{y}}\nabla_{\mathbf{y}}F_{t,w}\left(\mathbf{x}^k_t,\mathbf{y}^k_t\right)\right)$$

Let $\tau_t$ be the number of times the gradient update is executed at the $t$-th iteration. Note that $\mathbf{x}^{\tau_t}_t = \mathbf{x}_{t+1}$ and $\mathbf{y}^{\tau_t}_t = \mathbf{y}_{t+1}$.

#### B.3.1 Supporting Lemmas

We present three key lemmas which are important step descent lemmas. In this section, we focus on a crucial quantity, $\delta^k_{t,w} = \left\|\mathbf{y}^\star_{t,w}\left(\mathbf{x}^k_t\right) - \mathbf{y}^k_t\right\|^2$, which are useful for the subsequent analysis. Throughout our analysis, we choose $\eta_{\mathbf{x}} = \frac{1}{8\kappa^3\ell}$ and $\eta_{\mathbf{y}} = \frac{1}{\ell}$.

**Lemma B.5.** *Denote $\tau_t$ the total iteration of inner-loop at round $t$ with $1 \leq t \leq T-1$. For convenience, let $\tau_T = 0$. For $0 \leq k \leq \tau_t - 1$, we have*

$$\Phi_{t,w}\left(\mathbf{x}^{k+1}_t\right) \leq \Phi_{t,w}\left(\mathbf{x}^k_t\right) - \left(\frac{\eta_{\mathbf{x}}}{2} - \eta_{\mathbf{x}}^2\kappa\ell\right)\left\|\nabla_{\mathbf{x}}F_{t,w}\left(\mathbf{x}^k_t,\mathbf{y}^k_t\right)\right\|^2 + \frac{\eta_{\mathbf{x}}\ell^2}{2}\delta^k_{t,w}. \tag{22}$$

*Proof.* Since $\Phi_{t,w}$ is $(\ell + \kappa\ell)$-smooth and $\ell + \kappa\ell \le 2\kappa\ell$, for any $x, x^+ \in \mathbb{R}^m$, we have

$$\Phi_{t,w}\left(\mathbf{x}^+\right) - \Phi_{t,w}\left(\mathbf{x}\right) - \left(\mathbf{x}^+ - \mathbf{x}\right)^\top \nabla\Phi_{t,w}\left(\mathbf{x}\right) \le \kappa\ell \left\|\mathbf{x}^+ - \mathbf{x}\right\|^2.$$

Plugging $\mathbf{x}^+ - \mathbf{x} = -\eta_{\mathbf{x}}\nabla_{\mathbf{x}}F_{t,w}\left(\mathbf{x},\mathbf{y}\right)$ yields that

$$\begin{aligned}
\Phi_{t,w}\left(\mathbf{x}^+\right) \le {}& \Phi_{t,w}\left(\mathbf{x}\right) - \eta_{\mathbf{x}}\left\|\nabla_{\mathbf{x}}F_{t,w}\left(\mathbf{x},\mathbf{y}\right)\right\|^2 + \eta_{\mathbf{x}}^2\kappa\ell\left\|\nabla_{\mathbf{x}}F_{t,w}\left(\mathbf{x},\mathbf{y}\right)\right\|^2 \\
& + \eta_{\mathbf{x}}\left(\nabla_{\mathbf{x}}F_{t,w}\left(\mathbf{x},\mathbf{y}\right) - \nabla\Phi_{t,w}\left(\mathbf{x}\right)\right)^\top \nabla_{\mathbf{x}}F_{t,w}\left(\mathbf{x},\mathbf{y}\right).
\end{aligned}$$

By Young's inequality, we have

$$\begin{aligned}
& \left(\nabla_{\mathbf{x}}F_{t,w}\left(\mathbf{x},\mathbf{y}\right) - \nabla\Phi_{t,w}\left(\mathbf{x}\right)\right)^\top \nabla_{\mathbf{x}}F_{t,w}\left(\mathbf{x},\mathbf{y}\right) \\
& \le \frac{\left\|\nabla_{\mathbf{x}}F_{t,w}\left(\mathbf{x},\mathbf{y}\right) - \nabla\Phi_{t,w}\left(\mathbf{x}\right)\right\|^2 + \left\|\nabla_{\mathbf{x}}F_{t,w}\left(\mathbf{x},\mathbf{y}\right)\right\|^2}{2}.
\end{aligned}$$

Since $\nabla\Phi_{t,w}\left(\mathbf{x}\right) = \nabla_{\mathbf{x}}F_{t,w}\left(\mathbf{x},\mathbf{y}^*_{t,w}(\mathbf{x})\right)$, we have

$$\left\|\nabla_{\mathbf{x}}F_{t,w}\left(\mathbf{x},\mathbf{y}\right) - \nabla\Phi_{t,w}\left(\mathbf{x}\right)\right\|^2 \le \ell^2\|\mathbf{y} - \mathbf{y}^*_{t,w}(\mathbf{x})\|^2.$$

Putting these pieces together, we obtain

$$\begin{aligned}
\Phi_{t,w}\left(\mathbf{x}^+\right) \le {}& \Phi_{t,w}\left(\mathbf{x}\right) - \left(\frac{\eta_{\mathbf{x}}}{2} - \eta_{\mathbf{x}}^2\kappa\ell\right)\left\|\nabla_{\mathbf{x}}F_{t,w}\left(\mathbf{x},\mathbf{y}\right)\right\|^2 \\
& + \frac{\eta_{\mathbf{x}}\ell^2}{2}\|\mathbf{y} - \mathbf{y}^*_{t,w}(\mathbf{x})\|^2
\end{aligned}$$

$\square$

**Lemma B.6.** *For any $t, k \ge 0$, the following statement holds true,*

$$\|\mathbf{y}_t^{k+1} - \mathbf{y}_t^k\|^2 \le (4 - \frac{2}{\kappa})\delta_{t,w}^k. \tag{23}$$

*Proof.* By Young's inequality, we have

$$\begin{aligned}
\|\mathbf{y}_t^{k+1} - \mathbf{y}_t^k\|^2 & \le 2\|\mathbf{y}_t^{k+1} - \mathbf{y}^*_{t,w}\left(\mathbf{x}_t^k\right)\|^2 + 2\|\mathbf{y}^*_{t,w}\left(\mathbf{x}_t^k\right) - \mathbf{y}_t^k\|^2 \\
& \le \left(2(1 - \frac{1}{\kappa}) + 2\right)\delta_{t,w}^k = (4 - \frac{2}{\kappa})\delta_{t,w}^k.
\end{aligned}$$

$\square$

**Lemma B.7.** *Let $\delta_{t,w}^k = \left\|\mathbf{y}^\star_{t,w}\left(\mathbf{x}_t^k\right) - \mathbf{y}_t^k\right\|^2$, the following statement holds true,*

$$\delta_{t,w}^k \le \left(1 - \frac{1}{2\kappa}\right)\delta_{t,w}^{k-1} + 2\kappa^3\eta_{\mathbf{x}}^2\|\nabla_{\mathbf{x}}F_{t,w}(\mathbf{x}_t^{k-1},\mathbf{y}_t^{k-1})\|^2.$$

*Proof.* Since $f_t(\mathbf{x}, \cdot)$ is $\mu$-strongly concave and $\eta_{\mathbf{y}} = 1/\ell$, we have

$$\|\mathbf{y}^\star_{t,w}\left(\mathbf{x}_t^{k-1}\right) - \mathbf{y}_t^k\|^2 \le (1 - \frac{1}{\kappa})\delta_{t,w}^{k-1}.$$

By Young's inequality, we have

$$\begin{aligned}
\delta_{t,w}^k & \le \left(1 + \frac{1}{2(\kappa - 1)}\right)\|\mathbf{y}^\star_{t,w}\left(\mathbf{x}_t^{k-1}\right) - \mathbf{y}_t^k\|^2 + (1 + 2(\kappa - 1))\|\mathbf{y}^\star_{t,w}\left(\mathbf{x}_t^k\right) - \mathbf{y}^\star_{t,w}\left(\mathbf{x}_t^{k-1}\right)\|^2 \\
& \le \left(\frac{2\kappa - 1}{2\kappa - 2}\right)\|\mathbf{y}^\star_{t,w}\left(\mathbf{x}_t^{k-1}\right) - \mathbf{y}_t^k\|^2 + 2\kappa\|\mathbf{y}^\star_{t,w}\left(\mathbf{x}_t^k\right) - \mathbf{y}^\star_{t,w}\left(\mathbf{x}_t^{k-1}\right)\|^2 \\
& \le \left(1 - \frac{1}{2\kappa}\right)\delta_{t,w}^{k-1} + 2\kappa\|\mathbf{y}^\star_{t,w}\left(\mathbf{x}_t^k\right) - \mathbf{y}^\star_{t,w}\left(\mathbf{x}_t^{k-1}\right)\|^2. \tag{24}
\end{aligned}$$

Since $\mathbf{y}_{t,w}^{\star}(\cdot)$ is $\kappa$-Lipschitz, we have

$$\|\mathbf{y}_{t,w}^{\star}\left(\mathbf{x}_t^k\right) - \mathbf{y}_{t,w}^{\star}\left(\mathbf{x}_t^{k-1}\right)\|^2 \leq 2\kappa^2\|\mathbf{x}_t^k - \mathbf{x}_t^{k-1}\|^2 = 2\kappa^2\eta_{\mathbf{x}}^2\|\nabla_{\mathbf{x}}F_{t,w}(\mathbf{x}_t^{k-1}, \mathbf{y}_t^{k-1})\|^2.$$

Thus, plug into eq. (24)

$$\delta_{t,w}^k \leq \left(1 - \frac{1}{2\kappa}\right)\delta_{t,w}^{k-1} + 2\kappa^3\eta_{\mathbf{x}}^2\|\nabla_{\mathbf{x}}F_{t,w}(\mathbf{x}_t^{k-1}, \mathbf{y}_t^{k-1})\|^2.$$

$\square$

## B.4 Proof of Theorem 2

*Proof of Theorem 2.* Denote $\gamma = 1 - \frac{1}{2\kappa}$, from Lemma B.7 and using telescoping we have

$$\delta_{t,w}^k \leq \gamma^k\delta_{t,w}^0 + 2\kappa^3\eta_{\mathbf{x}}^2\left(\sum_{j=0}^{k-1}\gamma^{k-1-j}\left\|\nabla_{\mathbf{x}}F_{t,w}\left(\mathbf{x}_t^j, \mathbf{y}_t^j\right)\right\|^2\right). \tag{25}$$

Specially, for $t > 1$,

$$\begin{aligned}
\delta_{t,w}^0 &= \|\mathbf{y}_t^0 - \mathbf{y}_{t,w}^*(\mathbf{x}_t^0)\|^2 \\
&\leq 2\|\mathbf{y}_{t-1}^{\tau_{t-1}} - \mathbf{y}_{t-1,w}^*(\mathbf{x}_{t-1}^{\tau_{t-1}})\|^2 + 2\|\mathbf{y}_{t-1,w}^*(\mathbf{x}_{t-1}^{\tau_{t-1}}) - \mathbf{y}_{t,w}^*(\mathbf{x}_{t-1}^{\tau_{t-1}})\|^2 \\
&\leq \frac{\delta^2}{\ell^2 w^2} + \frac{2}{\mu^2(w-1)^2}\|\nabla_{\mathbf{y}}f_t(\mathbf{x}_{t-1}^{\tau_{t-1}}, \mathbf{y}_{t,w}^*(\mathbf{x}_{t-1}^{\tau_{t-1}})) - \nabla_{\mathbf{y}}f_{t-w}(\mathbf{x}_{t-1}^{\tau_{t-1}}, \mathbf{y}_{t-1,w}^*(\mathbf{x}_{t-1}^{\tau_{t-1}}))\|^2.
\end{aligned}$$

Then plug Equation (25) into Equations (22) and (23) from Lemmas B.5 and B.6, and sum over outer loop number.

$$\left(\frac{\eta_{\mathbf{x}}}{2} - \eta_{\mathbf{x}}^2\kappa\ell - 2\kappa^4\eta_{\mathbf{x}}^3\ell^2\right)\sum_{j=0}^{\tau_t-1}\left\|\nabla_{\mathbf{x}}F_{t,w}\left(\mathbf{x}_t^j, \mathbf{y}_t^j\right)\right\|^2 \leq \Phi_{t,w}\left(\mathbf{x}_t\right) - \Phi_{t,w}\left(\mathbf{x}_{t+1}\right) + \kappa\eta_{\mathbf{x}}\ell^2\delta_{t,w}^0$$

$$\sum_{j=0}^{\tau_t-1}\|\mathbf{y}_t^{k+1} - \mathbf{y}_t^k\|^2 \leq (8\kappa - 4)\delta_{t,w}^0 + (16 - \frac{8}{\kappa})\kappa^4\eta_{\mathbf{x}}^2\sum_{j=0}^{\tau_t-1}\left\|\nabla_{\mathbf{x}}F_{t,w}\left(\mathbf{x}_t^j, \mathbf{y}_t^j\right)\right\|^2.$$

Letting $\eta_{\mathbf{x}} = \frac{1}{8\kappa^3\ell}$, we have

$$\sum_{j=0}^{\tau_t-1}\left\|\nabla_{\mathbf{x}}F_{t,w}\left(\mathbf{x}_t^j, \mathbf{y}_t^j\right)\right\|^2 \leq \frac{8}{\eta_{\mathbf{x}}}(\Phi_{t,w}\left(\mathbf{x}_t\right) - \Phi_{t,w}\left(\mathbf{x}_{t+1}\right)) + 8\kappa\ell^2\delta_{t,w}^0 \tag{26}$$

$$\sum_{j=0}^{\tau_t-1}(\kappa\ell)^2\|\mathbf{y}_t^{j+1} - \mathbf{y}_t^j\|^2 \leq (8\kappa - 4)(\kappa\ell)^2\delta_{t,w}^0 + \frac{1}{4}\sum_{j=0}^{\tau_t-1}\left\|\nabla_{\mathbf{x}}F_{t,w}\left(\mathbf{x}_t^j, \mathbf{y}_t^j\right)\right\|^2. \tag{27}$$

Therefore add Equation (26) $\times\frac{5\eta_{\mathbf{x}}}{4}$ and Equation (27) $\times\frac{9\eta_{\mathbf{x}}}{2}$ we have

$$\frac{\eta_{\mathbf{x}}}{8}\sum_{j=0}^{\tau_t-1}\left[\left\|\nabla_{\mathbf{x}}F_{t,w}\left(\mathbf{x}_t^j, \mathbf{y}_t^j\right)\right\|^2 + (6\kappa\ell)^2\|\mathbf{y}_t^{j+1} - \mathbf{y}_t^j\|^2\right] \leq (\Phi_{t,w}\left(\mathbf{x}_t\right) - \Phi_{t,w}\left(\mathbf{x}_{t+1}\right)) + \frac{9\ell}{2}\delta_{t,w}^0. \tag{28}$$

Denote $\Phi_{0,w}(\mathbf{x}) = 0$, we notice that

$$\Phi_{T,w}(\mathbf{x}_T) = \sum_{t=1}^{T}(\Phi_{t,w}(\mathbf{x}_t) - \Phi_{t-1,w}(\mathbf{x}_{t-1}))$$

$$= \sum_{t=1}^{T}(\Phi_{t,w}(\mathbf{x}_t) - \Phi_{t-1,w}(\mathbf{x}_t)) + \sum_{t=2}^{T}(\Phi_{t-1,w}(\mathbf{x}_t) - \Phi_{t-1,w}(\mathbf{x}_{t-1}))$$

$$= \frac{1}{w}\sum_{t=1}^{T}\left(F_{t-1,w}(\mathbf{x}_t, \mathbf{y}_{t,w}^*(\mathbf{x}_t)) - F_{t-1,w}(\mathbf{x}_t, \mathbf{y}_{t-1,w}^*(\mathbf{x}_t))\right)$$

$$+ \frac{1}{w}\sum_{t=1}^{T}\left(f_t(\mathbf{x}_t, \mathbf{y}_{t,w}^*(\mathbf{x}_t)) - f_{t-w}(\mathbf{x}_t, \mathbf{y}_{t,w}^*(\mathbf{x}_t))\right) + \sum_{t=2}^{T}(\Phi_{t-1,w}(\mathbf{x}_t) - \Phi_{t-1,w}(\mathbf{x}_{t-1}))$$

$$\overset{(i)}{\leq} \frac{1}{w}\sum_{t=1}^{T}\left(f_t(\mathbf{x}_t, \mathbf{y}_{t,w}^*(\mathbf{x}_t)) - f_{t-w}(\mathbf{x}_t, \mathbf{y}_{t,w}^*(\mathbf{x}_t))\right) + \sum_{t=2}^{T}(\Phi_{t-1,w}(\mathbf{x}_t) - \Phi_{t-1,w}(\mathbf{x}_{t-1})),$$

where $(i)$ follows from that $\mathbf{y}_{t-1,w}^*(\mathbf{x}_t)$ is the maximizer of $F_{t-1,w}(\mathbf{x}_t, \cdot)$.

By some algebra, we have

$$\sum_{t=1}^{T-1}\Phi_{t,w}(\mathbf{x}_t) - \Phi_{t,w}(\mathbf{x}_{t+1}) \leq \frac{1}{w}\sum_{t=1}^{T}\left(f_t(\mathbf{x}_t, \mathbf{y}_{t,w}^*(\mathbf{x}_t)) - f_{t-w}(\mathbf{x}_t, \mathbf{y}_{t,w}^*(\mathbf{x}_t))\right) - \Phi_{T,w}(\mathbf{x}_T).$$

Sum Equation (28) over $t$, we have

$$\frac{\eta_{\mathbf{x}}}{8} \times \frac{\delta^2}{2w^2}\tau = \frac{\eta_{\mathbf{x}}\delta^2\tau}{16w^2}$$

$$\leq \frac{\eta_{\mathbf{x}}}{8}\sum_{t=1}^{T}\sum_{j=0}^{\tau_t-1}\left[\left\|\nabla_{\mathbf{x}}F_{t,w}\left(\mathbf{x}_t^j, \mathbf{y}_t^j\right)\right\|^2 + (\kappa\ell)^2\|\mathbf{y}_t^{j+1} - \mathbf{y}_t^j\|^2\right]$$

$$\leq \sum_{t=1}^{T}(\Phi_{t,w}(\mathbf{x}_t) - \Phi_{t,w}(\mathbf{x}_{t+1})) + \frac{9\ell}{2}\sum_{t=1}^{T}\delta_{t,w}^0$$

$$\leq \frac{1}{w}\sum_{t=1}^{T}\left(f_t(\mathbf{x}_t, \mathbf{y}_{t,w}^*(\mathbf{x}_t)) - f_{t-w}(\mathbf{x}_t, \mathbf{y}_{t,w}^*(\mathbf{x}_t))\right) - \Phi_{T+1,w}(\mathbf{x}_{T+1})$$

$$+ \frac{9T\delta^2}{2\ell w^2} + \frac{9\ell}{\mu^2(w-1)^2}V_w^2[T] + \frac{9\ell D^2}{2}$$

$$\leq \frac{2MT}{w} + M + \frac{9T\delta^2}{2\ell w^2} + \frac{9\ell}{\mu^2(w-1)^2}V_w^2[T] + \frac{9\ell D^2}{2}.$$

Hence

$$\tau \leq \frac{384\kappa^3\ell M w T}{\delta^2} + 576\frac{\kappa^2 T}{\mu} + 1152\frac{w^2\kappa^5}{(w-1)^2\delta^2}V_w^2[T] + \frac{576D^2\kappa^3\ell^2 w^2}{\delta^2}.$$

$\square$

## C  Missing Proof of Section 5

We first make some notation in Algorithm 2 clearly here, $G_t^{k+1} = \frac{1}{w}\sum_{i=t-w+1}^{t}\tilde{\nabla}f_i(\mathbf{x}_t^{k+1}, \mathbf{y}_t^{k+1}) = \tilde{\nabla}F_{t,w}(\mathbf{x}_t^{k+1}, \mathbf{y}_t^{k+1}) = \left(\tilde{\nabla}_{\mathbf{x}}F_{t,w}(\mathbf{x}_t^{k+1}, \mathbf{y}_t^{k+1}), \tilde{\nabla}_{\mathbf{y}}F_{t,w}(\mathbf{x}_t^{k+1}, \mathbf{y}_t^{k+1})\right)$. And for casimplification, we denote $\mathbf{y}_t^k = \mathbf{y}_t^{\tau_t}$ for any $k \geq \tau_t$.

Before our theoretical analysis of Algorithm 2 and proof of Section 5, we define the filtration in Algorithm 2 formally to describe clearly what is known and what is unknown at certain stage.

**Definition 4** (Filtration). *For any $t \geq 1$, we denote filtration $\mathcal{F}_t$ to be the $\sigma$-fields that corresponds to the randomness of all gradient feedback up to stage $t-1$ and the decision of $f_t$ at stage $t$. In particular, $\mathcal{F}_t$ includes $f_t, \mathbf{x}_t$ and $\tilde{\nabla} F_{t-1,w}(\mathbf{x}_t, \mathbf{y}_t)$, but doesn't include $\tilde{\nabla} f_t(\mathbf{x}_t, \mathbf{y}_t), \tilde{\nabla} F_{t,w}(\mathbf{x}_t, \mathbf{y}_t)$.*
*For any $t \geq 1$, $k \geq 1$, we denote filtration $\mathcal{F}_t^k$ to be the $\sigma$-fields that corresponds to the randomness of all gradient feedback up to the $k$-th iteration in line 6 at stage $t$ in Algorithm 2. In particular, $\mathcal{F}_t^k$ includes $f_t, \mathbf{x}_t^k, \mathbf{y}_t^k, \tilde{\nabla} F_{t,w}(\mathbf{x}_t, \mathbf{y}_t), \{\tilde{\nabla} f_i(\mathbf{x}_t^{k-1}, \mathbf{y}_t^{k-1})\}_{i=t-w+1}^t$ and $G_t^{k-1}$, but doesn't include $G_t^k, \{\tilde{\nabla} f_i(\mathbf{x}_t^k, \mathbf{y}_t^k)\}_{i=t-w+1}^t$.*

### C.1 Finite Iteration: Proof of Theorem 3

### C.1.1 Supporting Lemmas

Generally speaking, the lemmas in this section extends lemmas in Appendix B.3.1 to noisy setting. We first provide a descend lemma for $\Phi_{t,w}(\mathbf{x})$ in each iteration of inner loop.

**Lemma C.1.** *Denote $\tau_t$ the total iteration of inner-loop at stage $t$ and $\delta_{t,w}^k = \left\| \mathbf{y}_{t,w}^\star(\mathbf{x}_t^k) - \mathbf{y}_t^k \right\|^2$, for $0 \leq k \leq \tau_t - 1$*

$$\Phi_{t,w}\left(\mathbf{x}_t^{k+1}\right) \leq \Phi_{t,w}\left(\mathbf{x}_t^k\right) - \left(\frac{\eta_{\mathbf{x}}}{2} - \eta_{\mathbf{x}}^2 \kappa \ell\right) \left\| \tilde{\nabla}_{\mathbf{x}} F_{t,w}\left(\mathbf{x}, \mathbf{y}\right) \right\|^2 + \eta_{\mathbf{x}} \ell^2 \delta_{t,w}^k$$
$$+ \left\| \tilde{\nabla}_{\mathbf{x}} F_{t,w}\left(\mathbf{x}, \mathbf{y}\right) - \nabla_{\mathbf{x}} F_{t,w}\left(\mathbf{x}, \mathbf{y}\right) \right\|^2.$$

*Proof.* Since $\Phi_{t,w}$ is $(\ell + \kappa \ell)$-smooth, for any $x, x^+ \in \mathbb{R}^m$, we have

$$\Phi_{t,w}\left(\mathbf{x}^+\right) - \Phi_{t,w}\left(\mathbf{x}\right) - \left(\mathbf{x}^+ - \mathbf{x}\right)^\top \nabla \Phi_{t,w}\left(\mathbf{x}\right) \leq \kappa \ell \left\| \mathbf{x}^+ - \mathbf{x} \right\|^2.$$

Set $\mathbf{x}^+ = \mathbf{x}_t^{k+1}, \mathbf{x} = \mathbf{x}_t^k$, we have $\mathbf{x}^+ - \mathbf{x} = \mathbf{x}_t^{k+1} - \mathbf{x}_t^k = -\eta_{\mathbf{x}} \tilde{\nabla}_{\mathbf{x}} F_{t,w}\left(\mathbf{x}_t^k, \mathbf{y}_t^k\right)$, which yeilds that

$$\Phi_{t,w}\left(\mathbf{x}_t^{k+1}\right) \leq \Phi_{t,w}\left(\mathbf{x}_t^k\right) - \eta_{\mathbf{x}} \left\| \tilde{\nabla}_{\mathbf{x}} F_{t,w}\left(\mathbf{x}_t^k, \mathbf{y}_t^k\right) \right\|^2 + \eta_{\mathbf{x}}^2 \kappa \ell \left\| \tilde{\nabla}_{\mathbf{x}} F_{t,w}\left(\mathbf{x}_t^k, \mathbf{y}_t^k\right) \right\|^2$$
$$+ \eta_{\mathbf{x}} \left(\tilde{\nabla}_{\mathbf{x}} F_{t,w}\left(\mathbf{x}_t^k, \mathbf{y}_t^k\right) - \nabla \Phi_{t,w}\left(\mathbf{x}_t^k\right)\right)^\top \tilde{\nabla}_{\mathbf{x}} F_{t,w}\left(\mathbf{x}_t^k, \mathbf{y}_t^k\right). \tag{29}$$

By Young's inequality, we have

$$\left(\tilde{\nabla}_{\mathbf{x}} F_{t,w}\left(\mathbf{x}_t^k, \mathbf{y}_t^k\right) - \nabla \Phi_{t,w}\left(\mathbf{x}_t^k\right)\right)^\top \tilde{\nabla}_{\mathbf{x}} F_{t,w}\left(\mathbf{x}_t^k, \mathbf{y}_t^k\right)$$
$$\leq \frac{\|\tilde{\nabla}_{\mathbf{x}} F_{t,w}\left(\mathbf{x}_t^k, \mathbf{y}_t^k\right) - \nabla \Phi_{t,w}\left(\mathbf{x}_t^k\right)\|^2 + \|\tilde{\nabla}_{\mathbf{x}} F_{t,w}\left(\mathbf{x}_t^k, \mathbf{y}_t^k\right)\|^2}{2}$$
$$\leq \frac{2\|\tilde{\nabla}_{\mathbf{x}} F_{t,w}\left(\mathbf{x}_t^k, \mathbf{y}_t^k\right) - \nabla_{\mathbf{x}} F_{t,w}\left(\mathbf{x}_t^k, \mathbf{y}_t^k\right)\|^2 + 2\|\nabla_{\mathbf{x}} F_{t,w}\left(\mathbf{x}_t^k, \mathbf{y}_t^k\right) - \nabla \Phi_{t,w}\left(\mathbf{x}_t^k\right)\|^2}{2}$$
$$+ \frac{\|\tilde{\nabla}_{\mathbf{x}} F_{t,w}\left(\mathbf{x}_t^k, \mathbf{y}_t^k\right)\|^2}{2}. \tag{30}$$

Since $\nabla \Phi_{t,w}\left(\mathbf{x}_t^k\right) = \nabla_{\mathbf{x}} F_{t,w}\left(\mathbf{x}_t^k, \mathbf{y}_{t,w}^*(\mathbf{x}_t^k)\right)$, we have

$$\|\nabla_{\mathbf{x}} F_{t,w}\left(\mathbf{x}_t^k, \mathbf{y}_t^k\right) - \nabla \Phi_{t,w}\left(\mathbf{x}_t^k\right)\|^2 \leq \ell^2 \|\mathbf{y}_t^k - \mathbf{y}_{t,w}^*(\mathbf{x}_t^k)\|^2. \tag{31}$$

Putting Equations (29) to (31) together, we obtain

$$\Phi_{t,w}\left(\mathbf{x}_t^{k+1}\right) \leq \Phi_{t,w}\left(\mathbf{x}_t^k\right) - \left(\frac{\eta_{\mathbf{x}}}{2} - \eta_{\mathbf{x}}^2 \kappa \ell\right) \left\| \tilde{\nabla}_{\mathbf{x}} F_{t,w}\left(\mathbf{x}_t^k, \mathbf{y}_t^k\right) \right\|^2$$
$$+ \eta_{\mathbf{x}} \ell^2 \|\mathbf{y}_t^k - \mathbf{y}_{t,w}^*(\mathbf{x}_t^k)\|^2 + \|\tilde{\nabla}_{\mathbf{x}} F_{t,w}\left(\mathbf{x}_t^k, \mathbf{y}_t^k\right) - \nabla_{\mathbf{x}} F_{t,w}\left(\mathbf{x}_t^k, \mathbf{y}_t^k\right)\|^2.$$

$\square$

The next lemma characterizes the descent property of distance to the maximizer $\mathbf{y}_{t,w}^{\star}$.

**Lemma C.2.** *Let* $\delta_{t,w}^{k} = \left\| \mathbf{y}_{t,w}^{\star}\left(\mathbf{x}_{t}^{k}\right) - \mathbf{y}_{t}^{k} \right\|^{2}$, *the following statement holds true,*

$$\delta_{t,w}^{k} \leq \left(1 - \frac{1}{4\kappa}\right)\delta_{t,w}^{k-1} + 8\kappa^{3}\eta_{\mathbf{x}}^{2}\|\tilde{\nabla}_{\mathbf{x}}F_{t,w}(\mathbf{x}_{t}^{k-1}, \mathbf{y}_{t}^{k-1})\|^{2}$$
$$+ \frac{2\kappa}{\ell^{2}}\left\|\nabla_{\mathbf{x}}F_{t,w}(\mathbf{x}_{t}^{k-1}, \mathbf{y}_{t}^{k-1}) - \tilde{\nabla}_{\mathbf{x}}F_{t,w}(\mathbf{x}_{t}^{k-1}, \mathbf{y}_{t}^{k-1})\right\|^{2}.$$

*Proof.* Since $f(\mathbf{x}, \cdot)$ is $\mu$-strongly concave and $\eta_{\mathbf{y}} = 1/\ell$, we have

$$\|\mathbf{y}_{t,w}^{\star}\left(\mathbf{x}_{t}^{k-1}\right) - \mathbf{y}_{t}^{k}\|^{2}$$
$$= \left\|\mathbf{y}_{t,w}^{\star}\left(\mathbf{x}_{t}^{k-1}\right) - \mathcal{P}_{\mathcal{Y}}\left(\mathbf{y}_{t}^{k-1} + \eta_{\mathbf{y}}\tilde{\nabla}_{\mathbf{x}}F_{t,w}(\mathbf{x}_{t}^{k-1}, \mathbf{y}_{t}^{k-1})\right)\right\|^{2}$$
$$= \left\|\mathbf{y}_{t,w}^{\star}\left(\mathbf{x}_{t}^{k-1}\right) - \mathcal{P}_{\mathcal{Y}}\left(\mathbf{y}_{t}^{k-1} + \eta_{\mathbf{y}}\nabla_{\mathbf{x}}F_{t,w}(\mathbf{x}_{t}^{k-1}, \mathbf{y}_{t}^{k-1})\right)\right.$$
$$\left. + \mathcal{P}_{\mathcal{Y}}\left(\mathbf{y}_{t}^{k-1} + \eta_{\mathbf{y}}\nabla_{\mathbf{x}}F_{t,w}(\mathbf{x}_{t}^{k-1}, \mathbf{y}_{t}^{k-1})\right) - \mathcal{P}_{\mathcal{Y}}\left(\mathbf{y}_{t}^{k-1} + \eta_{\mathbf{y}}\tilde{\nabla}_{\mathbf{x}}F_{t,w}(\mathbf{x}_{t}^{k-1}, \mathbf{y}_{t}^{k-1})\right)\right\|^{2}$$
$$\leq (1 + \frac{1}{2(\kappa - 1)})\left\|\mathbf{y}_{t,w}^{\star}\left(\mathbf{x}_{t}^{k-1}\right) - \mathcal{P}_{\mathcal{Y}}\left(\mathbf{y}_{t}^{k-1} + \eta_{\mathbf{y}}\nabla_{\mathbf{x}}F_{t,w}(\mathbf{x}_{t}^{k-1}, \mathbf{y}_{t}^{k-1})\right)\right\|^{2}$$
$$+ (1 + 2(\kappa - 1))\left\|\mathcal{P}_{\mathcal{Y}}\left(\mathbf{y}_{t}^{k-1} + \eta_{\mathbf{y}}\nabla_{\mathbf{x}}F_{t,w}(\mathbf{x}_{t}^{k-1}, \mathbf{y}_{t}^{k-1})\right) - \mathcal{P}_{\mathcal{Y}}\left(\mathbf{y}_{t}^{k-1} + \eta_{\mathbf{y}}\tilde{\nabla}_{\mathbf{x}}F_{t,w}(\mathbf{x}_{t}^{k-1}, \mathbf{y}_{t}^{k-1})\right)\right\|^{2}$$
$$\leq (1 - \frac{1}{2\kappa})\delta_{t,w}^{k-1} + \frac{2\kappa - 1}{\ell^{2}}\left\|\nabla_{\mathbf{x}}F_{t,w}(\mathbf{x}_{t}^{k-1}, \mathbf{y}_{t}^{k-1}) - \tilde{\nabla}_{\mathbf{x}}F_{t,w}(\mathbf{x}_{t}^{k-1}, \mathbf{y}_{t}^{k-1})\right\|^{2}. \tag{32}$$

By Young's inequality, we have

$$\delta_{t,w}^{k} \leq \left(1 + \frac{1}{2(2\kappa - 1)}\right)\|\mathbf{y}_{t,w}^{\star}\left(\mathbf{x}_{t}^{k-1}\right) - \mathbf{y}_{t}^{k}\|^{2}$$
$$+ (1 + 2(2\kappa - 1))\|\mathbf{y}_{t,w}^{\star}\left(\mathbf{x}_{t}^{k}\right) - \mathbf{y}_{t,w}^{\star}\left(\mathbf{x}_{t}^{k-1}\right)\|^{2}$$
$$\leq \left(\frac{4\kappa - 1}{2(2\kappa - 1)}\right)\|\mathbf{y}_{t,w}^{\star}\left(\mathbf{x}_{t}^{k-1}\right) - \mathbf{y}_{t}^{k}\|^{2} + 4\kappa\|\mathbf{y}_{t,w}^{\star}\left(\mathbf{x}_{t}^{k}\right) - \mathbf{y}_{t,w}^{\star}\left(\mathbf{x}_{t}^{k-1}\right)\|^{2}$$
$$\leq \left(1 - \frac{1}{4\kappa}\right)\delta_{t,w}^{k-1} + 4\kappa\|\mathbf{y}_{t,w}^{\star}\left(\mathbf{x}_{t}^{k}\right) - \mathbf{y}_{t,w}^{\star}\left(\mathbf{x}_{t}^{k-1}\right)\|^{2}$$
$$+ \frac{2\kappa}{\ell^{2}}\left\|\nabla_{\mathbf{x}}F_{t,w}(\mathbf{x}_{t}^{k-1}, \mathbf{y}_{t}^{k-1}) - \tilde{\nabla}_{\mathbf{x}}F_{t,w}(\mathbf{x}_{t}^{k-1}, \mathbf{y}_{t}^{k-1})\right\|^{2}.$$

Since $\mathbf{y}_{t,w}^{\star}(\cdot)$ is $\kappa$-Lipschitz, we have

$$\|\mathbf{y}_{t,w}^{\star}\left(\mathbf{x}_{t}^{k}\right) - \mathbf{y}_{t,w}^{\star}\left(\mathbf{x}_{t}^{k-1}\right)\|^{2} \leq 2\kappa^{2}\|\mathbf{x}_{t}^{k} - \mathbf{x}_{t}^{k-1}\|^{2}$$
$$= 2\kappa^{2}\eta_{\mathbf{x}}^{2}\|\tilde{\nabla}_{\mathbf{x}}F_{t,w}(\mathbf{x}_{t}^{k-1}, \mathbf{y}_{t}^{k-1})\|^{2}.$$

Thus, plug into

$$\delta_{t,w}^{k} \leq \left(1 - \frac{1}{4\kappa}\right)\delta_{t,w}^{k-1} + 8\kappa^{3}\eta_{\mathbf{x}}^{2}\|\tilde{\nabla}_{\mathbf{x}}F_{t,w}(\mathbf{x}_{t}^{k-1}, \mathbf{y}_{t}^{k-1})\|^{2}$$
$$+ \frac{2\kappa}{\ell^{2}}\left\|\nabla_{\mathbf{x}}F_{t,w}(\mathbf{x}_{t}^{k-1}, \mathbf{y}_{t}^{k-1}) - \tilde{\nabla}_{\mathbf{x}}F_{t,w}(\mathbf{x}_{t}^{k-1}, \mathbf{y}_{t}^{k-1})\right\|^{2}.$$

$\square$

The next lemma shows that updates over $\mathbf{y}$ can be controlled by $\delta_{t,w}^{k}$ plus a noisy term.

**Lemma C.3.** *For any* $t, k \geq 0$, *the following statement holds true,*

$$\|\mathbf{y}_{t}^{k+1} - \mathbf{y}_{t}^{k}\|^{2} \leq (4 - \frac{1}{\kappa})\delta_{t,w}^{k} + \frac{4\kappa}{\ell^{2}}\left\|\nabla_{\mathbf{x}}F_{t,w}(\mathbf{x}_{t}^{k}, \mathbf{y}_{t}^{k}) - \tilde{\nabla}_{\mathbf{x}}F_{t,w}(\mathbf{x}_{t}^{k}, \mathbf{y}_{t}^{k})\right\|^{2}.$$

*Proof.* By Young's inequality, we have

$$
\begin{aligned}
\|\mathbf{y}_t^{k+1} - \mathbf{y}_t^k\|^2 &\leq 2\|\mathbf{y}_t^{k+1} - \mathbf{y}_{t,w}^\star\left(\mathbf{x}_t^k\right)\|^2 + 2\|\mathbf{y}_{t,w}^\star\left(\mathbf{x}_t^k\right) - \mathbf{y}_t^k\|^2 \\
&\overset{(i)}{\leq} \left(2(1 - \frac{1}{2\kappa}) + 2\right)\delta_{t,w}^k + \frac{4\kappa}{\ell^2}\left\|\nabla_{\mathbf{x}}F_{t,w}(\mathbf{x}_t^k, \mathbf{y}_t^k) - \tilde{\nabla}_{\mathbf{x}}F_{t,w}(\mathbf{x}_t^k, \mathbf{y}_t^k)\right\|^2 \\
&\leq (4 - \frac{1}{\kappa})\delta_{t,w}^k + \frac{4\kappa}{\ell^2}\left\|\nabla_{\mathbf{x}}F_{t,w}(\mathbf{x}_t^k, \mathbf{y}_t^k) - \tilde{\nabla}_{\mathbf{x}}F_{t,w}(\mathbf{x}_t^k, \mathbf{y}_t^k)\right\|^2,
\end{aligned}
$$

where $(i)$ follows from Equation (32). $\qquad\square$

### C.1.2  Proof of Theorem 3

*Proof.* From Lemma C.1

$$
\begin{aligned}
\delta_{t,w}^k &\leq \left(1 - \frac{1}{4\kappa}\right)\delta_{t,w}^{k-1} + 8\kappa^3\eta_{\mathbf{x}}^2\|\tilde{\nabla}_{\mathbf{x}}F_{t,w}(\mathbf{x}_t^{k-1}, \mathbf{y}_t^{k-1})\|^2 \\
&\quad + \frac{2\kappa}{\ell^2}\left\|\nabla_{\mathbf{x}}F_{t,w}(\mathbf{x}_t^{k-1}, \mathbf{y}_t^{k-1}) - \tilde{\nabla}_{\mathbf{x}}F_{t,w}(\mathbf{x}_t^{k-1}, \mathbf{y}_t^{k-1})\right\|^2.
\end{aligned}
$$

Denote $\gamma = 1 - \frac{1}{4\kappa}$, Given $\mathcal{F}_t^{k-1}$ we have

$$
\begin{aligned}
\delta_{t,w}^k &\leq \gamma^k\delta_{t,w}^0 + 8\kappa^3\eta_{\mathbf{x}}^2\left(\sum_{j=0}^{k-1}\gamma^{k-1-j}\left\|\tilde{\nabla}_{\mathbf{x}}F_{t,w}\left(\mathbf{x}_t^j, \mathbf{y}_t^j\right)\right\|^2\right) \\
&\quad + \frac{2\kappa}{\ell^2}\left(\sum_{j=0}^{k-1}\gamma^{k-1-j}\left\|\tilde{\nabla}_{\mathbf{x}}F_{t,w}(\mathbf{x}_t^j, \mathbf{y}_t^j) - \nabla_{\mathbf{x}}F_{t,w}(\mathbf{x}_t^j, \mathbf{y}_t^j)\right\|^2\right) \\
&\overset{(i)}{\leq} \gamma^k D^2 + \frac{32\kappa^4\eta_{\mathbf{x}}^2\delta^2}{3w^2} + \frac{2\kappa}{\ell^2}\left(\sum_{j=0}^{k-1}\gamma^{k-1-j}\left\|\tilde{\nabla}_{\mathbf{x}}F_{t,w}(\mathbf{x}_t^j, \mathbf{y}_t^j) - \nabla_{\mathbf{x}}F_{t,w}(\mathbf{x}_t^j, \mathbf{y}_t^j)\right\|^2\right),
\end{aligned}
\tag{33}
$$

where the first term of $(i)$ follows from that $\mathcal{Y}$ is bounded with $D$, and the second term of $(i)$ follows from the stopping criterion of Algorithm 2 and $\sum_{j=0}^{k-1}\gamma^{k-1-j} \leq 4\kappa$.

Notice that for any fixed $t, k$ and $j \in [k-1]$,

$$
\begin{aligned}
\mathbb{E}&\left\|\tilde{\nabla}_{\mathbf{x}}F_{t,w}(\mathbf{x}_t^j, \mathbf{y}_t^j) - \nabla_{\mathbf{x}}F_{t,w}(\mathbf{x}_t^j, \mathbf{y}_t^j)\right\|^2 \\
&\overset{(i)}{=} \mathbb{E}_{\mathcal{F}_t^j}\left[\mathbb{E}\left[\left\|\tilde{\nabla}_{\mathbf{x}}F_{t,w}(\mathbf{x}_t^j, \mathbf{y}_t^j) - \nabla_{\mathbf{x}}F_{t,w}(\mathbf{x}_t^j, \mathbf{y}_t^j)\right\|^2\Big|\mathcal{F}_t^j\right]\right] \\
&= \mathbb{E}_{\mathcal{F}_t^j}\left[\frac{1}{w^2}\mathbb{E}\left[\left\|\sum_{i=t-w}^{t-1}\left\{\tilde{\nabla}_{\mathbf{x}}f_i(\mathbf{x}_t^j, \mathbf{y}_t^j) - \nabla_{\mathbf{x}}f_i(\mathbf{x}_t^j, \mathbf{y}_t^j)\right\}\right\|^2\Big|\mathcal{F}_t^j\right]\right] \\
&\overset{(ii)}{=} \mathbb{E}_{\mathcal{F}_t^j}\left[\frac{1}{w^2}\sum_{i=t-w}^{t-1}\mathbb{E}\left[\left\|\tilde{\nabla}_{\mathbf{x}}f_i(\mathbf{x}_t^j, \mathbf{y}_t^j) - \nabla_{\mathbf{x}}f_i(\mathbf{x}_t^j, \mathbf{y}_t^j)\right\|^2\Big|\mathcal{F}_t^j\right]\right] \\
&\overset{(iii)}{=} \mathbb{E}_{\mathcal{F}_t^j}\left[\frac{1}{w^2}\cdot w\cdot\frac{\sigma^2}{w^2}\right] = \frac{\sigma^2}{w^3},
\end{aligned}
\tag{34}
$$

where $(i)$ follows from the property of conditional expectation, $(ii)$ follows from that the SFO calls in line 9 of Algorithm 2 is independent and $(iii)$ follows from definition of SFO and filtration $\mathcal{F}_t^j$.

Thus take expectation over two sides of Equation (33), we have

$$
\mathbb{E}\left[\delta_{t,w}^k\right] \leq \gamma^k D^2 + \frac{32\kappa^4\eta_{\mathbf{x}}^2\delta^2}{3w^2} + \frac{8\kappa^2\sigma^2}{\ell^2 w^3}.
\tag{35}
$$

Then by Lemma C.2

$$
\begin{aligned}
\Phi_{t,w}&\left(\mathbf{x}_t^k\right) - \Phi_{t,w}\left(\mathbf{x}_t^{k+1}\right) \\
&\geq (\frac{\eta_{\mathbf{x}}}{2} - \eta_{\mathbf{x}}^2\kappa\ell)\left\|\tilde{\nabla}_{\mathbf{x}}F_{t,w}\left(\mathbf{x}_t^k,\mathbf{y}_t^k\right)\right\|^2 \\
&\quad - \eta_{\mathbf{x}}\ell^2\delta_{t,w}^k - \|\tilde{\nabla}_{\mathbf{x}}F_{t,w}\left(\mathbf{x}_t^k,\mathbf{y}_t^k\right) - \nabla_{\mathbf{x}}F_{t,w}\left(\mathbf{x}_t^k,\mathbf{y}_t^k\right)\|^2 \\
&\geq \frac{15\eta_{\mathbf{x}}}{32}\left\|\tilde{\nabla}_{\mathbf{x}}F_{t,w}\left(\mathbf{x}_t^k,\mathbf{y}_t^k\right)\right\|^2 \\
&\quad - \eta_{\mathbf{x}}\ell^2\delta_{t,w}^k - \|\tilde{\nabla}_{\mathbf{x}}F_{t,w}\left(\mathbf{x}_t^k,\mathbf{y}_t^k\right) - \nabla_{\mathbf{x}}F_{t,w}\left(\mathbf{x}_t^k,\mathbf{y}_t^k\right)\|^2.
\end{aligned}
\tag{36}
$$

By Lemma C.3

$$
\frac{15}{4}\kappa^2\ell^2\eta_{\mathbf{x}}\delta_{t,w}^k + \frac{15}{4\ell}\left\|\nabla_{\mathbf{x}}F_{t,w}(\mathbf{x}_t^k,\mathbf{y}_t^k) - \tilde{\nabla}_{\mathbf{x}}F_{t,w}(\mathbf{x}_t^k,\mathbf{y}_t^k)\right\|^2 \geq \frac{15\eta_{\mathbf{x}}}{32}\times 2\kappa^2\ell^2\|\mathbf{y}_t^{k+1}-\mathbf{y}_t^k\|^2.
\tag{37}
$$

Sum Equation (36) and Equation (37), we have

$$
\begin{aligned}
\Phi_{t,w}&\left(\mathbf{x}_t^k\right) - \Phi_{t,w}\left(\mathbf{x}_t^{k+1}\right) + \frac{15}{4}\kappa^2\ell^2\eta_{\mathbf{x}}\delta_{t,w}^k + \frac{15}{4\ell}\left\|\nabla_{\mathbf{x}}F_{t,w}(\mathbf{x}_t^k,\mathbf{y}_t^k) - \tilde{\nabla}_{\mathbf{x}}F_{t,w}(\mathbf{x}_t^k,\mathbf{y}_t^k)\right\|^2 \\
&\geq \frac{15\eta_{\mathbf{x}}}{32}\times\left(2\kappa^2\ell^2\|\mathbf{y}_t^{k+1}-\mathbf{y}_t^k\|^2 + \left\|\tilde{\nabla}_{\mathbf{x}}F_{t,w}\left(\mathbf{x}_t^k,\mathbf{y}_t^k\right)\right\|^2\right) \\
&\quad - \eta_{\mathbf{x}}\ell^2\delta_{t,w}^k - \|\tilde{\nabla}_{\mathbf{x}}F_{t,w}\left(\mathbf{x}_t^k,\mathbf{y}_t^k\right) - \nabla_{\mathbf{x}}F_{t,w}\left(\mathbf{x}_t^k,\mathbf{y}_t^k\right)\|^2
\end{aligned}
$$

Rearranging the term, we have

$$
\begin{aligned}
\Phi_{t,w}&\left(\mathbf{x}_t^k\right) - \Phi_{t,w}\left(\mathbf{x}_t^{k+1}\right) \\
&\geq \frac{15\eta_{\mathbf{x}}}{32}\times\left(2\kappa^2\ell^2\|\mathbf{y}_t^{k+1}-\mathbf{y}_t^k\|^2 + \left\|\tilde{\nabla}_{\mathbf{x}}F_{t,w}\left(\mathbf{x}_t^k,\mathbf{y}_t^k\right)\right\|^2\right) \\
&\quad - 5\kappa^2\ell^2\eta_{\mathbf{x}}\delta_{t,w}^k - \left(\frac{15}{4\ell}+1\right)\|\tilde{\nabla}_{\mathbf{x}}F_{t,w}\left(\mathbf{x}_t^k,\mathbf{y}_t^k\right) - \nabla_{\mathbf{x}}F_{t,w}\left(\mathbf{x}_t^k,\mathbf{y}_t^k\right)\|^2.
\end{aligned}
\tag{38}
$$

Take expectation over both sides of Equation (38), plug into Equation (35) and follow from the similar step of Equation (34), we have

$$
\begin{aligned}
\mathbb{E}&\left[\Phi_{t,w}\left(\mathbf{x}_t^k\right) - \Phi_{t,w}\left(\mathbf{x}_t^{k+1}\right)\right] \\
&\geq \frac{5\eta_{\mathbf{x}}\delta^2}{32w^2} - 5\kappa^2\ell^2\eta_{\mathbf{x}}\left(\gamma^k D^2 + \frac{32\kappa^4\eta_{\mathbf{x}}^2\delta^2}{3w^2} + \frac{8\kappa^2\sigma^2}{\ell^2 w^3}\right) - \left(\frac{15}{4\ell}+1\right)\frac{\sigma^2}{w^3}.
\end{aligned}
$$

Because $\gamma = 1 - \frac{1}{4\kappa} \leq 1$, there exist a constant $\tilde{K}$ such that $\gamma^{\tilde{K}}D^2 \leq \max\left\{\frac{32\kappa^4\eta_{\mathbf{x}}^2}{3w^2}, \frac{8\kappa^2\sigma^2}{\ell^2 w^3}\right\}$. Thus for $k \geq \tilde{K}$, we have

$$
\begin{aligned}
\mathbb{E}&\left[\Phi_{t,w}\left(\mathbf{x}_t^k\right) - \Phi_{t,w}\left(\mathbf{x}_t^{k+1}\right)\right] \\
&\geq \frac{5\eta_{\mathbf{x}}\delta^2}{32w^2} - 5\kappa^2\ell^2\eta_{\mathbf{x}}\left(\frac{35\kappa^4\eta_{\mathbf{x}}^2\delta^2}{3w^2} + \frac{9\kappa^2\sigma^2}{\ell^2 w^3}\right) - \left(\frac{15}{4\ell}+1\right)\frac{\sigma^2}{w^3} \\
&\geq \frac{25\eta_{\mathbf{x}}\delta^2}{256w^2} - \frac{45\kappa^4\eta_{\mathbf{x}}\sigma^2}{w^3} - \left(\frac{15}{4\ell}+1\right)\frac{\sigma^2}{w^3}.
\end{aligned}
$$

when $\delta^2 > \frac{2304\kappa^4\sigma^2}{5w} + \frac{256(4\ell+1)\sigma^2}{25\eta_{\mathbf{x}}w}$, we set $\alpha = \frac{25\eta_{\mathbf{x}}\delta^2}{256w^2} - \frac{45\kappa^4\eta_{\mathbf{x}}\sigma^2}{w^3} - \left(\frac{15}{4\ell}+1\right)\frac{\sigma^2}{w^3} > 0$. Then for $K \geq \tilde{K}$, we have

$$
\begin{aligned}
2M &\geq \mathbb{E}\left[\Phi_{t,w}\left(\mathbf{x}_t^{\tilde{K}}\right) - \Phi_{t,w}\left(\mathbf{x}_t^{K+1}\right)\right] \\
&= \mathbb{E}\left[\sum_{k=\tilde{K}}^{K}\left(\Phi_{t,w}\left(\mathbf{x}_t^k\right) - \Phi_{t,w}\left(\mathbf{x}_t^{k+1}\right)\right)\right] \\
&= \sum_{k=\tilde{K}}^{K}\left(\mathbb{E}\left[\left(\Phi_{t,w}\left(\mathbf{x}_t^k\right) - \Phi_{t,w}\left(\mathbf{x}_t^{k+1}\right)\right)\middle|\tau_t \geq k+1\right]\mathbb{P}\left(\tau_t \geq k+1\right) + 0\cdot\mathbb{P}\left(\tau_t < k+1\right)\right) \\
&\geq \alpha\sum_{k=\tilde{K}}^{K}\mathbb{P}\left(\tau_t \geq k+1\right) \\
&\geq \alpha\sum_{k=\tilde{K}}^{K}\mathbb{P}\left(\tau_t > K\right) = \alpha\left(K-\tilde{K}\right)\mathbb{P}\left(\tau_t > K\right),
\end{aligned}
$$

where the third equation follows from the Optional Stopping Theorem. Consequently, we have $\tau_t$ is finite in probability, which implies that $\tau = \sum_{t=1}^{T}\tau_t$ is finite in probability since it is the finite sum of finite variables in probability. $\qquad\square$

## C.2    Local Regret: Proof of Theorem 4

*Proof of Theorem 4.* Following from Equation (21), we have

$$
\begin{aligned}
\|\nabla\Phi_{t,w}\left(\mathbf{x}_t\right)\|^2 &= \left\|\nabla_{\mathbf{x}}F_{t,w}(\mathbf{x}_t,\mathbf{y}_{t,w}^*(\mathbf{x}_t))\right\|^2 \\
&\leq 3\left\|\nabla\Phi_{t-1,w}\left(\mathbf{x}_t\right)\right\|^2 + \frac{3\kappa^2}{(w-1)^2}\|\nabla_{\mathbf{y}}f_t\left(\mathbf{x}_t,\mathbf{y}_{t,w}^*(\mathbf{x}_t)\right) - \nabla_{\mathbf{y}}f_{t-w}\left(\mathbf{x}_t,\mathbf{y}_{t-1,w}^*(\mathbf{x}_t)\right)\|^2 \\
&\quad + \frac{3}{w^2}\|\nabla_{\mathbf{x}}f_t\left(\mathbf{x}_t,\mathbf{y}_{t,w}^*(\mathbf{x}_t)\right) - \nabla_{\mathbf{x}}f_{t-w}\left(\mathbf{x}_t,\mathbf{y}_{t,w}^*(\mathbf{x}_t)\right)\|^2.
\end{aligned}
\tag{39}
$$

For the first term

$$
\begin{aligned}
&\|\nabla\Phi_{t-1,w}(\mathbf{x}_t)\|^2 \\
&= \left\|\nabla\Phi_{t-1,w}(\mathbf{x}_{t-1}^{\tau_{t-1}})\right\|^2 \\
&\leq 3\left\|\nabla\Phi_{t-1,w}(\mathbf{x}_{t-1}^{\tau_{t-1}}) - \nabla_{\mathbf{x}}F_{t-1,w}(\mathbf{x}_{t-1}^{\tau_{t-1}},\mathbf{y}_{t-1}^{\tau_{t-1}})\right\|^2 \\
&\quad + 3\left\|\nabla_{\mathbf{x}}F_{t-1,w}(\mathbf{x}_{t-1}^{\tau_{t-1}},\mathbf{y}_{t-1}^{\tau_{t-1}}) - \tilde{\nabla}_{\mathbf{x}}F_{t-1,w}(\mathbf{x}_{t-1}^{\tau_{t-1}},\mathbf{y}_{t-1}^{\tau_{t-1}})\right\|^2 + 3\left\|\tilde{\nabla}_{\mathbf{x}}F_{t-1,w}(\mathbf{x}_{t-1}^{\tau_{t-1}},\mathbf{y}_{t-1}^{\tau_{t-1}})\right\|^2 \\
&\leq 3\ell^2\left\|\mathbf{y}_{t-1}^\star(\mathbf{x}_{t-1}^{\tau_{t-1}}) - \mathbf{y}_{t-1}^{\tau_{t-1}}\right\|^2 + 3\left\|\tilde{\nabla}_{\mathbf{x}}F_{t-1,w}(\mathbf{x}_{t-1}^{\tau_{t-1}},\mathbf{y}_{t-1}^{\tau_{t-1}})\right\|^2 \\
&\quad + 3\left\|\nabla_{\mathbf{x}}F_{t-1,w}(\mathbf{x}_{t-1}^{\tau_{t-1}},\mathbf{y}_{t-1}^{\tau_{t-1}}) - \tilde{\nabla}_{\mathbf{x}}F_{t-1,w}(\mathbf{x}_{t-1}^{\tau_{t-1}},\mathbf{y}_{t-1}^{\tau_{t-1}})\right\|^2.
\end{aligned}
$$

Consider $\left\| \mathbf{y}_{t-1}^{\star}(\mathbf{x}_{t-1}^{\tau_{t-1}}) - \mathbf{y}_{t-1}^{\tau_{t-1}} \right\|^2$

$$
\begin{aligned}
&\left\| \mathbf{y}_{t-1}^{\star}(\mathbf{x}_{t-1}^{\tau_{t-1}}) - \mathbf{y}_{t-1}^{\tau_{t-1}} \right\|^2 \\
&\overset{(i)}{\leq} (\frac{2\kappa}{\ell\eta_{\mathbf{y}}} + 1)^2 (1 + \ell\eta_{\mathbf{y}})^2 \left\| \mathbf{y}_{t-1}^{\tau_{t-1}} - \mathcal{P}_{\mathcal{Y}} \left( \mathbf{y}_t + \eta_{\mathbf{y}} \nabla_{\mathbf{y}} F_{t-1,w}(\mathbf{x}_{t-1}^{\tau_{t-1}}, \mathbf{y}_{t-1}^{\tau_{t-1}}) \right) \right\|^2 \\
&\leq 2(\frac{2\kappa}{\ell\eta_{\mathbf{y}}} + 1)^2 (1 + \ell\eta_{\mathbf{y}})^2 \left\| \mathbf{y}_{t-1}^{\tau_{t-1}} - \mathcal{P}_{\mathcal{Y}} \left( \mathbf{y}_t + \eta_{\mathbf{y}} \tilde{\nabla}_{\mathbf{y}} F_{t-1,w}(\mathbf{x}_{t-1}^{\tau_{t-1}}, \mathbf{y}_{t-1}^{\tau_{t-1}}) \right) \right\|^2 \\
&\quad + 2(\frac{2\kappa}{\ell\eta_{\mathbf{y}}} + 1)^2 (1 + \ell\eta_{\mathbf{y}})^2 \left\| \mathcal{P}_{\mathcal{Y}} \left( \mathbf{y}_t + \eta_{\mathbf{y}} \nabla_{\mathbf{y}} F_{t-1,w}(\mathbf{x}_{t-1}^{\tau_{t-1}}, \mathbf{y}_{t-1}^{\tau_{t-1}}) \right) - \mathcal{P}_{\mathcal{Y}} \left( \mathbf{y}_t + \eta_{\mathbf{y}} \tilde{\nabla}_{\mathbf{y}} F_{t-1,w}(\mathbf{x}_{t-1}^{\tau_{t-1}}, \mathbf{y}_{t-1}^{\tau_{t-1}}) \right) \right\|^2 \\
&\overset{(ii)}{\leq} 2(\frac{2\kappa}{\ell\eta_{\mathbf{y}}} + 1)^2 (1 + \ell\eta_{\mathbf{y}})^2 \left\| \mathbf{y}_{t-1}^{\tau_{t-1}} - \mathcal{P}_{\mathcal{Y}} \left( \mathbf{y}_t + \eta_{\mathbf{y}} \tilde{\nabla}_{\mathbf{y}} F_{t-1,w}(\mathbf{x}_{t-1}^{\tau_{t-1}}, \mathbf{y}_{t-1}^{\tau_{t-1}}) \right) \right\|^2 \\
&\quad + 2(\frac{2\kappa}{\ell\eta_{\mathbf{y}}} + 1)^2 (1 + \ell\eta_{\mathbf{y}})^2 \cdot \eta_{\mathbf{y}}^2 \left\| \nabla_{\mathbf{y}} F_{t-1,w}(\mathbf{x}_{t-1}^{\tau_{t-1}}, \mathbf{y}_{t-1}^{\tau_{t-1}}) - \tilde{\nabla}_{\mathbf{y}} F_{t-1,w}(\mathbf{x}_{t-1}^{\tau_{t-1}}, \mathbf{y}_{t-1}^{\tau_{t-1}}) \right\|^2 ,
\end{aligned}
$$

where $(i)$ follows from Lemma B.4 similar to Lemma B.3 and $(ii)$ follows from the project operator is a contraction.

Then

$$
\begin{aligned}
&\left\| \nabla \Phi_{t-1,w}(\mathbf{x}_t) \right\|^2 \\
&\leq 6(\frac{2\kappa}{\eta_{\mathbf{y}}} + \ell)^2 (1 + \ell\eta_{\mathbf{y}})^2 \left\| \mathbf{y}_{t-1}^{\tau_{t-1}} - \mathcal{P}_{\mathcal{Y}} \left( \mathbf{y}_t + \eta_{\mathbf{y}} \tilde{\nabla}_{\mathbf{y}} F_{t-1,w}(\mathbf{x}_{t-1}^{\tau_{t-1}}, \mathbf{y}_{t-1}^{\tau_{t-1}}) \right) \right\|^2 + 3 \left\| \tilde{\nabla}_{\mathbf{x}} F_{t-1,w}(\mathbf{x}_{t-1}^{\tau_{t-1}}, \mathbf{y}_{t-1}^{\tau_{t-1}}) \right\|^2 \\
&\quad + 6(2\kappa + \ell\eta_{\mathbf{y}})^2 (1 + \ell\eta_{\mathbf{y}})^2 \left\| \nabla_{\mathbf{y}} F_{t-1,w}(\mathbf{x}_{t-1}^{\tau_{t-1}}, \mathbf{y}_{t-1}^{\tau_{t-1}}) - \tilde{\nabla}_{\mathbf{y}} F_{t-1,w}(\mathbf{x}_{t-1}^{\tau_{t-1}}, \mathbf{y}_{t-1}^{\tau_{t-1}}) \right\|^2 \\
&\quad + 3 \left\| \nabla_{\mathbf{x}} F_{t-1,w}(\mathbf{x}_{t-1}^{\tau_{t-1}}, \mathbf{y}_{t-1}^{\tau_{t-1}}) - \tilde{\nabla}_{\mathbf{x}} F_{t-1,w}(\mathbf{x}_{t-1}^{\tau_{t-1}}, \mathbf{y}_{t-1}^{\tau_{t-1}}) \right\|^2 \\
&\overset{(i)}{\leq} \frac{\delta^2}{w^2} + 6(2\kappa + \ell\eta_{\mathbf{y}})^2 (1 + \ell\eta_{\mathbf{y}})^2 \left\| \nabla_{\mathbf{y}} F_{t-1,w}(\mathbf{x}_{t-1}^{\tau_{t-1}}, \mathbf{y}_{t-1}^{\tau_{t-1}}) - \tilde{\nabla}_{\mathbf{y}} F_{t-1,w}(\mathbf{x}_{t-1}^{\tau_{t-1}}, \mathbf{y}_{t-1}^{\tau_{t-1}}) \right\|^2 \\
&\quad + 3 \left\| \nabla_{\mathbf{x}} F_{t-1,w}(\mathbf{x}_{t-1}^{\tau_{t-1}}, \mathbf{y}_{t-1}^{\tau_{t-1}}) - \tilde{\nabla}_{\mathbf{x}} F_{t-1,w}(\mathbf{x}_{t-1}^{\tau_{t-1}}, \mathbf{y}_{t-1}^{\tau_{t-1}}) \right\|^2 ,
\end{aligned}
\tag{40}
$$

where $(i)$ follows from Stop Condition 2 of inner-loop.

Plug Equation (40) into Equation (39) and sum over $t$, we have

$$
\begin{aligned}
R_w(T) &= \sum_{t=1}^{T} \left\| \nabla \Phi_{t,w}(\mathbf{x}_t) \right\|^2 \\
&\leq \sum_{t=1}^{T} \Bigg\{ \frac{3\delta^2}{w^2} + 18(2\kappa + \ell\eta_{\mathbf{y}})^2 (1 + \ell\eta_{\mathbf{y}})^2 \left\| \nabla_{\mathbf{y}} F_{t-1,w}(\mathbf{x}_{t-1}^{\tau_{t-1}}, \mathbf{y}_{t-1}^{\tau_{t-1}}) - \tilde{\nabla}_{\mathbf{y}} F_{t-1,w}(\mathbf{x}_{t-1}^{\tau_{t-1}}, \mathbf{y}_{t-1}^{\tau_{t-1}}) \right\|^2 \\
&\quad + 9 \left\| \nabla_{\mathbf{x}} F_{t-1,w}(\mathbf{x}_{t-1}^{\tau_{t-1}}, \mathbf{y}_{t-1}^{\tau_{t-1}}) - \tilde{\nabla}_{\mathbf{x}} F_{t-1,w}(\mathbf{x}_{t-1}^{\tau_{t-1}}, \mathbf{y}_{t-1}^{\tau_{t-1}}) \right\|^2 \\
&\quad + \frac{3\kappa^2}{(w-1)^2} \| \nabla_{\mathbf{y}} f_t \left( \mathbf{x}_{t-1}^{\tau_{t-1}}, \mathbf{y}_{t,w}^*(\mathbf{x}_{t-1}^{\tau_{t-1}}) \right) - \nabla_{\mathbf{y}} f_{t-w} \left( \mathbf{x}_{t-1}^{\tau_{t-1}}, \mathbf{y}_{t-1,w}^*(\mathbf{x}_{t-1}^{\tau_{t-1}}) \right) \|^2 \\
&\quad + \frac{3}{w^2} \left\| \nabla_{\mathbf{x}} f_t \left( \mathbf{x}_{t-1}^{\tau_{t-1}}, \mathbf{y}_{t,w}^*(\mathbf{x}_{t-1}^{\tau_{t-1}}) \right) - \nabla_{\mathbf{x}} f_{t-w} \left( \mathbf{x}_{t-1}^{\tau_{t-1}}, \mathbf{y}_{t,w}^*(\mathbf{x}_{t-1}^{\tau_{t-1}}) \right) \right\|^2 \Bigg\} \\
&= \frac{3T\delta^2}{w^2} + \frac{3\kappa^2}{(w-1)^2} V_w^2[T] + \frac{3}{w^2} V_w^1[T] \\
&\quad + \sum_{t=1}^{T} \Bigg\{ 18(2\kappa + \ell\eta_{\mathbf{y}})^2 (1 + \ell\eta_{\mathbf{y}})^2 \left\| \nabla_{\mathbf{y}} F_{t-1,w}(\mathbf{x}_{t-1}^{\tau_{t-1}}, \mathbf{y}_{t-1}^{\tau_{t-1}}) - \tilde{\nabla}_{\mathbf{y}} F_{t-1,w}(\mathbf{x}_{t-1}^{\tau_{t-1}}, \mathbf{y}_{t-1}^{\tau_{t-1}}) \right\|^2 \\
&\quad + 9 \left\| \nabla_{\mathbf{x}} F_{t-1,w}(\mathbf{x}_{t-1}^{\tau_{t-1}}, \mathbf{y}_{t-1}^{\tau_{t-1}}) - \tilde{\nabla}_{\mathbf{x}} F_{t-1,w}(\mathbf{x}_{t-1}^{\tau_{t-1}}, \mathbf{y}_{t-1}^{\tau_{t-1}}) \right\|^2 \Bigg\} .
\end{aligned}
\tag{41}
$$

Notice that for any $t \in [T]$,

$$
\begin{aligned}
&\mathbb{E}\big\|\nabla_{\mathbf{y}}F_{t-1,w}(\mathbf{x}_{t-1}^{\tau_{t-1}},\mathbf{y}_{t-1}^{\tau_{t-1}}) - \tilde{\nabla}_{\mathbf{y}}F_{t-1,w}(\mathbf{x}_{t-1}^{\tau_{t-1}},\mathbf{y}_{t-1}^{\tau_{t-1}})\big\|^2 \\
&\overset{(i)}{=} \mathbb{E}_{\mathcal{F}_{t-1}^{\tau_{t-1}}}\left[\mathbb{E}\left[\big\|\nabla_{\mathbf{y}}F_{t-1,w}(\mathbf{x}_{t-1}^{\tau_{t-1}},\mathbf{y}_{t-1}^{\tau_{t-1}}) - \tilde{\nabla}_{\mathbf{y}}F_{t-1,w}(\mathbf{x}_{t-1}^{\tau_{t-1}},\mathbf{y}_{t-1}^{\tau_{t-1}})\big\|^2\Big|\mathcal{F}_{t-1}^{\tau_{t-1}}\right]\right] \\
&= \mathbb{E}_{\mathcal{F}_{t-1}^{\tau_{t-1}}}\left[\frac{1}{w^2}\mathbb{E}\left[\Big\|\sum_{i=t-w}^{t-1}\big\{\nabla_{\mathbf{y}}f_i(\mathbf{x}_{t-1}^{\tau_{t-1}},\mathbf{y}_{t-1}^{\tau_{t-1}}) - \tilde{\nabla}_{\mathbf{y}}f_i(\mathbf{x}_{t-1}^{\tau_{t-1}},\mathbf{y}_{t-1}^{\tau_{t-1}})\big\}\Big\|^2\Big|\mathcal{F}_{t-1}^{\tau_{t-1}}\right]\right] \\
&\overset{(ii)}{=} \mathbb{E}_{\mathcal{F}_{t-1}^{\tau_{t-1}}}\left[\frac{1}{w^2}\sum_{i=t-w}^{t-1}\mathbb{E}\left[\big\|\nabla_{\mathbf{y}}f_i(\mathbf{x}_{t-1}^{\tau_{t-1}},\mathbf{y}_{t-1}^{\tau_{t-1}}) - \tilde{\nabla}_{\mathbf{y}}f_i(\mathbf{x}_{t-1}^{\tau_{t-1}},\mathbf{y}_{t-1}^{\tau_{t-1}})\big\|^2\Big|\mathcal{F}_{t-1}^{\tau_{t-1}}\right]\right] \\
&\overset{(iii)}{=} \mathbb{E}_{\mathcal{F}_{t-1}^{\tau_{t-1}}}\left[\frac{1}{w^2}\cdot w\cdot\frac{\sigma^2}{w^2}\right] = \frac{\sigma^2}{w^3},
\end{aligned}
\tag{42}
$$

where $(i)$ follows from the property of conditional expectation, $(ii)$ follows from that the SFO calls in line 9 of Algorithm 2 is independent and $(iii)$ follows from definition of SFO.

Similarly, for any $t$, we have

$$
\mathbb{E}\big\|\nabla_{\mathbf{x}}F_{t-1,w}(\mathbf{x}_{t-1}^{\tau_{t-1}},\mathbf{y}_{t-1}^{\tau_{t-1}}) - \tilde{\nabla}_{\mathbf{x}}F_{t-1,w}(\mathbf{x}_{t-1}^{\tau_{t-1}},\mathbf{y}_{t-1}^{\tau_{t-1}})\big\|^2 = \frac{\sigma^2}{w^3}.
\tag{43}
$$

Plug Equations (42) and (43) into Equation (41), we have

$$
\begin{aligned}
\mathbb{E}\left[\Re_w^{NE}(T)\right] &= \sum_{t=1}^{T}\mathbb{E}\left[\|\nabla\Phi_{t,w}(\mathbf{x}_t)\|^2\right] \\
&\leq \frac{3T\delta^2}{w^2} + \frac{3\kappa^2}{(w-1)^2}V_w^2[T] + \frac{3}{w^2}V_w^1[T] + \frac{\left(18(2\kappa+\ell\eta_{\mathbf{y}})^2(1+\ell\eta_{\mathbf{y}})^2+9\right)T\sigma^2}{w^3} \\
&\leq \frac{3T\delta^2}{w^2} + \frac{3\kappa^2}{(w-1)^2}V_w^2[T] + \frac{3}{w^2}V_w^1[T] + \frac{\left(360\kappa^2+9\right)T\sigma^2}{w^3}.
\end{aligned}
$$

$\square$

## C.3 Iteration and SFO Calls Bound: Proof of Theorem 5

*Proof of Theorem 5.* From Lemma C.1

$$
\begin{aligned}
\delta_{t,w}^k &\leq \left(1-\frac{1}{4\kappa}\right)\delta_{t,w}^{k-1} + 8\kappa^3\eta_{\mathbf{x}}^2\|\tilde{\nabla}_{\mathbf{x}}F_{t,w}(\mathbf{x}_t^{k-1},\mathbf{y}_t^{k-1})\|^2 \\
&\quad + \frac{2\kappa}{\ell^2}\left\|\nabla_{\mathbf{x}}F_{t,w}(\mathbf{x}_t^{k-1},\mathbf{y}_t^{k-1}) - \tilde{\nabla}_{\mathbf{x}}F_{t,w}(\mathbf{x}_t^{k-1},\mathbf{y}_t^{k-1})\right\|^2.
\end{aligned}
$$

Denote $\gamma = 1 - \frac{1}{4\kappa}$, Given $\mathcal{F}_t$ we have

$$
\begin{aligned}
\delta_{t,w}^k &\leq \gamma^k\delta_{t,w}^0 + 8\kappa^3\eta_{\mathbf{x}}^2\left(\sum_{j=0}^{k-1}\gamma^{k-1-j}\left\|\tilde{\nabla}_{\mathbf{x}}F_{t,w}\left(\mathbf{x}_t^j,\mathbf{y}_t^j\right)\right\|^2\right) \\
&\quad + \frac{2\kappa}{\ell^2}\left(\sum_{j=0}^{k-1}\gamma^{k-1-j}\left\|\tilde{\nabla}_{\mathbf{x}}F_{t,w}(\mathbf{x}_t^j,\mathbf{y}_t^j) - \nabla_{\mathbf{x}}F_{t,w}(\mathbf{x}_t^j,\mathbf{y}_t^j)\right\|^2\right).
\end{aligned}
\tag{44}
$$

Then by Lemma C.2

$$
\begin{aligned}
\Phi_{t,w}\left(\mathbf{x}_t^{k+1}\right) &\leq \Phi_{t,w}\left(\mathbf{x}_t^k\right) - \left(\frac{\eta_{\mathbf{x}}}{2} - \eta_{\mathbf{x}}^2\kappa\ell\right)\left\|\tilde{\nabla}_{\mathbf{x}}F_{t,w}\left(\mathbf{x}_t^k,\mathbf{y}_t^k\right)\right\|^2 \\
&\quad + \eta_{\mathbf{x}}\ell^2\delta_{t,w}^k + \|\tilde{\nabla}_{\mathbf{x}}F_{t,w}\left(\mathbf{x}_t^k,\mathbf{y}_t^k\right) - \nabla_{\mathbf{x}}F_{t,w}\left(\mathbf{x}_t^k,\mathbf{y}_t^k\right)\|^2.
\end{aligned}
\tag{45}
$$

Then plugging Equation (44) into Equation (45) and summing up them over $k = 0, \ldots, \tau_t - 1$, we have

$$
\begin{aligned}
\Phi_{t,w}(\mathbf{x}_t^{\tau_t}) \leq{} & \Phi_{t,w}(\mathbf{x}_t^0) - (\frac{\eta_{\mathbf{x}}}{2} - \eta_{\mathbf{x}}^2 \kappa \ell) \sum_{k=0}^{\tau_t - 1} \left\| \tilde{\nabla}_{\mathbf{x}} F_{t,w}\left(\mathbf{x}_t^k, \mathbf{y}_t^k\right) \right\|^2 + \eta_{\mathbf{x}} \ell^2 \sum_{k=0}^{\tau_t - 1} \gamma^k \delta_{t,w}^0 \\
& + 8\kappa^3 \eta_{\mathbf{x}}^3 \ell^2 \sum_{k=0}^{\tau_t - 1} \left( \sum_{j=0}^{k-1} \gamma^{k-1-j} \left\| \tilde{\nabla}_{\mathbf{x}} F_{t,w}\left(\mathbf{x}_t^j, \mathbf{y}_t^j\right) \right\|^2 \right) \\
& + 2\eta_{\mathbf{x}} \kappa \sum_{k=0}^{\tau_t - 1} \left( \sum_{j=0}^{k-1} \gamma^{k-1-j} \left\| \tilde{\nabla}_{\mathbf{x}} F_{t,w}(\mathbf{x}_t^j, \mathbf{y}_t^j) - \nabla_{\mathbf{x}} F_{t,w}(\mathbf{x}_t^j, \mathbf{y}_t^j) \right\|^2 \right) \\
& + \sum_{k=0}^{\tau_t - 1} \left\| \tilde{\nabla}_{\mathbf{x}} F_{t,w}\left(\mathbf{x}_t^k, \mathbf{y}_t^k\right) - \nabla_{\mathbf{x}} F_{t,w}\left(\mathbf{x}_t^k, \mathbf{y}_t^k\right) \right\|^2 \\
\leq{} & \Phi_{t,w}(\mathbf{x}_t^0) - (\frac{\eta_{\mathbf{x}}}{2} - \eta_{\mathbf{x}}^2 \kappa \ell - 32\kappa^4 \eta_{\mathbf{x}}^3 \ell^2) \sum_{k=0}^{\tau_t - 1} \left\| \tilde{\nabla}_{\mathbf{x}} F_{t,w}\left(\mathbf{x}_t^k, \mathbf{y}_t^k\right) \right\|^2 \\
& + 4\kappa \eta_{\mathbf{x}} \ell^2 \delta_{t,w}^0 + \left(8\kappa^2 \eta_{\mathbf{x}} + 1\right) \left( \sum_{k=0}^{\tau_t - 1} \left\| \tilde{\nabla}_{\mathbf{x}} F_{t,w}(\mathbf{x}_t^k, \mathbf{y}_t^k) - \nabla_{\mathbf{x}} F_{t,w}(\mathbf{x}_t^k, \mathbf{y}_t^k) \right\|^2 \right),
\end{aligned}
$$

where the last inequality follows from that $\sum_{k=0}^{\tau_t - 1} \gamma^k = \frac{1 - \gamma^{\tau_t}}{1 - \gamma} \leq 4\kappa$ and changing the order of summation over $j$ and $k$.

Rearranging the terms, we have

$$
\begin{aligned}
(\frac{\eta_{\mathbf{x}}}{2} &- \eta_{\mathbf{x}}^2 \kappa \ell - 32\kappa^4 \eta_{\mathbf{x}}^3 \ell^2) \sum_{k=0}^{\tau_t - 1} \left\| \tilde{\nabla}_{\mathbf{x}} F_{t,w}\left(\mathbf{x}_t^k, \mathbf{y}_t^k\right) \right\|^2 \\
&\leq \Phi_{t,w}\left(\mathbf{x}_t\right) - \Phi_{t,w}\left(\mathbf{x}_{t+1}\right) + 4\kappa \eta_{\mathbf{x}} \ell^2 \delta_{t,w}^0 \\
&\quad + \left(8\kappa^2 \eta_{\mathbf{x}} + 1\right) \left( \sum_{k=0}^{\tau_t - 1} \left\| \tilde{\nabla}_{\mathbf{x}} F_{t,w}(\mathbf{x}_t^k, \mathbf{y}_t^k) - \nabla_{\mathbf{x}} F_{t,w}(\mathbf{x}_t^k, \mathbf{y}_t^k) \right\|^2 \right).
\end{aligned}
$$

By Lemma C.3

$$
\|\mathbf{y}_t^{k+1} - \mathbf{y}_t^k\|^2 \leq (4 - \frac{1}{\kappa})\delta_{t,w}^k + \frac{4\kappa}{\ell^2} \left\| \nabla_{\mathbf{x}} F_{t,w}(\mathbf{x}_t^k, \mathbf{y}_t^k) - \tilde{\nabla}_{\mathbf{x}} F_{t,w}(\mathbf{x}_t^k, \mathbf{y}_t^k) \right\|^2.
$$

Then

$$
\begin{aligned}
\sum_{k=0}^{\tau_t - 1} \|\mathbf{y}_t^{k+1} - \mathbf{y}_t^k\|^2 \leq{} & (16\kappa - 4)\delta_{t,w}^0 + 128\kappa^4 \eta_{\mathbf{x}}^2 \sum_{k=0}^{\tau_t - 1} \left\| \tilde{\nabla}_{\mathbf{x}} F_{t,w}\left(\mathbf{x}_t^k, \mathbf{y}_t^k\right) \right\|^2 \\
& + \frac{36\kappa^2}{\ell^2} \sum_{k=0}^{\tau_t - 1} \left\| \tilde{\nabla}_{\mathbf{x}} F_{t,w}(\mathbf{x}_t^k, \mathbf{y}_t^k) - \nabla_{\mathbf{x}} F_{t,w}(\mathbf{x}_t^k, \mathbf{y}_t^k) \right\|^2.
\end{aligned}
$$

Notice that $\delta_{0,w}^0 \leq D^2$ and for any $t \geq 2$

$$
\begin{aligned}
\delta_{t,w}^0 &= \|\mathbf{y}_t^0 - \mathbf{y}_{t,w}^*(\mathbf{x}_t^0)\|^2 \\
&\leq 2\|\mathbf{y}_{t-1}^{\tau_{t-1}} - \mathbf{y}_{t-1,w}^*(\mathbf{x}_{t-1}^{\tau_{t-1}})\|^2 + 2\|\mathbf{y}_{t-1,w}^*(\mathbf{x}_{t-1}^{\tau_{t-1}}) - \mathbf{y}_{t,w}^*(\mathbf{x}_{t-1}^{\tau_{t-1}})\|^2 \\
&\leq 2\kappa^2 \|\mathbf{y}_{t-1}^{\tau_{t-1}} - \mathcal{P}_{\mathcal{Y}}\left(\mathbf{y}_{t-1}^{\tau_{t-1}} + \eta_{\mathbf{y}} G_{\mathbf{y},t-1}^{\tau_{t-1}}\right)\|^2 \\
&\quad + \frac{2}{\mu^2(w-1)^2}\|\nabla_{\mathbf{y}} f_t(\mathbf{x}_{t-1}^{\tau_{t-1}}, \mathbf{y}_{t,w}^*(\mathbf{x}_{t-1}^{\tau_{t-1}})) - \nabla_{\mathbf{y}} f_{t-w}(\mathbf{x}_{t-1}^{\tau_{t-1}}, \mathbf{y}_{t-1,w}^*(\mathbf{x}_{t-1}^{\tau_{t-1}}))\|^2 \\
&\leq \frac{\delta^2}{4\ell^2 w^2} + \frac{2}{\mu^2(w-1)^2}\|\nabla_{\mathbf{y}} f_t(\mathbf{x}_{t-1}^{\tau_{t-1}}, \mathbf{y}_{t,w}^*(\mathbf{x}_{t-1}^{\tau_{t-1}})) - \nabla_{\mathbf{y}} f_{t-w}(\mathbf{x}_{t-1}^{\tau_{t-1}}, \mathbf{y}_{t-1,w}^*(\mathbf{x}_{t-1}^{\tau_{t-1}}))\|^2.
\end{aligned}
$$

Letting $\eta_{\mathbf{x}} = \frac{1}{32\kappa^3\ell}$ and $\eta_{\mathbf{y}} = \frac{1}{\ell}$, we have

$$\sum_{k=0}^{\tau_t-1} \left\|\tilde{\nabla}_{\mathbf{x}} F_{t,w}\left(\mathbf{x}_t^k, \mathbf{y}_t^k\right)\right\|^2 \leq \frac{16}{7\eta_{\mathbf{x}}}(\Phi_{t,w}\left(\mathbf{x}_t^k\right) - \Phi_{t,w}\left(\mathbf{x}_{t+1}^k\right)) + \frac{64\kappa\ell^2\delta_{t,w}^0}{7}$$

$$+ \frac{640}{7\eta_{\mathbf{x}}}\left(\sum_{k=0}^{\tau_t-1}\left\|\tilde{\nabla}_{\mathbf{x}} F_{t,w}(\mathbf{x}_t^k, \mathbf{y}_t^k) - \nabla_{\mathbf{x}} F_{t,w}(\mathbf{x}_t^k, \mathbf{y}_t^k)\right\|^2\right) \qquad (46)$$

$$\sum_{k=0}^{\tau_t-1} 2\left(\left(\frac{2\kappa}{\eta_{\mathbf{y}}} + \ell\right)(1 + \ell\eta_{\mathbf{y}})\right)^2 \|\mathbf{y}_t^{k+1} - \mathbf{y}_t^k\|^2 = \sum_{k=0}^{\tau_t-1} 72(\kappa\ell)^2\|\mathbf{y}_t^{k+1} - \mathbf{y}_t^k\|^2$$

$$\leq 72(\kappa\ell)^2(16\kappa - 4)\delta_{t,w}^0 + 9\sum_{k=0}^{\tau_t-1}\left\|\tilde{\nabla}_{\mathbf{x}} F_{t,w}\left(\mathbf{x}_t^k, \mathbf{y}_t^k\right)\right\|^2$$

$$+ 2596\kappa^4\left(\sum_{k=0}^{\tau_t-1}\left\|\tilde{\nabla}_{\mathbf{x}} F_{t,w}(\mathbf{x}_t^k, \mathbf{y}_t^k) - \nabla_{\mathbf{x}} F_{t,w}(\mathbf{x}_t^k, \mathbf{y}_t^k)\right\|^2\right). \qquad (47)$$

Therefore add Equation (46) $\times 10\eta_{\mathbf{x}}$ and Equation (47) $\times \eta_{\mathbf{x}}$, we have

$$\eta_{\mathbf{x}}\sum_{k=0}^{\tau_t-1}\left[\left\|\tilde{\nabla}_{\mathbf{x}} F_{t,w}\left(\mathbf{x}_t^k, \mathbf{y}_t^k\right)\right\|^2 + 2\left(\left(\frac{2\kappa}{\eta_{\mathbf{y}}} + \ell\right)(1 + \ell\eta_{\mathbf{y}})\right)^2 \|\mathbf{y}_t^{k+1} - \mathbf{y}_t^k\|^2\right]$$

$$\leq \frac{160}{7}(\Phi_{t,w}\left(\mathbf{x}_t\right) - \Phi_{t,w}\left(\mathbf{x}_{t+1}\right)) + 1152\eta_{\mathbf{x}}\kappa^3\ell^2\delta_{t,w}^0$$

$$+ 2688\eta_{\mathbf{x}}\kappa^4\left(\sum_{k=0}^{\tau_t-1}\left\|\tilde{\nabla}_{\mathbf{x}} F_{t,w}(\mathbf{x}_t^k, \mathbf{y}_t^k) - \nabla_{\mathbf{x}} F_{t,w}(\mathbf{x}_t^k, \mathbf{y}_t^k)\right\|^2\right). \qquad (48)$$

Denote $\Phi_{0,w}(\mathbf{x}) = 0$, we notice that

$$\Phi_{T,w}(\mathbf{x}_T)$$

$$= \sum_{t=1}^{T}(\Phi_{t,w}(\mathbf{x}_t) - \Phi_{t-1,w}(\mathbf{x}_{t-1}))$$

$$= \sum_{t=1}^{T}(\Phi_{t,w}(\mathbf{x}_t) - \Phi_{t-1,w}(\mathbf{x}_t)) + \sum_{t=2}^{T}(\Phi_{t-1,w}(\mathbf{x}_t) - \Phi_{t-1,w}(\mathbf{x}_{t-1}))$$

$$= \frac{1}{w}\sum_{t=1}^{T}\left(F_{t-1,w}(\mathbf{x}_t, \mathbf{y}_{t,w}^*(\mathbf{x}_t)) - F_{t-1,w}(\mathbf{x}_t, \mathbf{y}_{t-1,w}^*(\mathbf{x}_t))\right)$$

$$+ \frac{1}{w}\sum_{t=1}^{T}\left(f_t(\mathbf{x}_t, \mathbf{y}_{t,w}^*(\mathbf{x}_t)) - f_{t-w}(\mathbf{x}_t, \mathbf{y}_{t,w}^*(\mathbf{x}_t))\right) + \sum_{t=2}^{T}(\Phi_{t-1,w}(\mathbf{x}_t) - \Phi_{t-1,w}(\mathbf{x}_{t-1}))$$

$$\overset{(i)}{\leq} \frac{1}{w}\sum_{t=1}^{T}\left(f_t(\mathbf{x}_t, \mathbf{y}_{t,w}^*(\mathbf{x}_t)) - f_{t-w}(\mathbf{x}_t, \mathbf{y}_{t,w}^*(\mathbf{x}_t))\right) + \sum_{t=2}^{T}(\Phi_{t-1,w}(\mathbf{x}_t) - \Phi_{t-1,w}(\mathbf{x}_{t-1})),$$

where $(i)$ follows from that $\mathbf{y}_{t-1,w}^*(\mathbf{x}_t)$ is the maximizer of $F_{t-1,w}(\mathbf{x}_t, \cdot)$.

By some algebra, we have

$$\sum_{t=1}^{T}\Phi_{t,w}(\mathbf{x}_t)) - (\Phi_{t,w}(\mathbf{x}_{t+1}) \leq \frac{1}{w}\sum_{t=1}^{T}\left(f_t(\mathbf{x}_t, \mathbf{y}_{t,w}^*(\mathbf{x}_t)) - f_{t-w}(\mathbf{x}_t, \mathbf{y}_{t,w}^*(\mathbf{x}_t))\right) - \Phi_{T+1,w}(\mathbf{x}_{T+1}).$$

Sum Equation ([48]) over $t$ and take expectation, we have

$$\left(\frac{\delta^2}{3w^2} - 2688\kappa^4\frac{\sigma^2}{w^2}\right)\eta_{\mathbf{x}}\tau$$

$$\leq \sum_{t=1}^{T}\eta_{\mathbf{x}}\sum_{k=0}^{\tau_t-1}\left[\left\|\tilde{\nabla}_{\mathbf{x}}F_{t,w}\left(\mathbf{x}_t^k,\mathbf{y}_t^k\right)\right\|^2 + 2\left((\frac{2\kappa}{\eta_{\mathbf{y}}} + \ell)(1 + \ell\eta_{\mathbf{y}})\right)^2\|\mathbf{y}_t^{k+1} - \mathbf{y}_t^k\|^2\right]$$

$$- 2688\eta_{\mathbf{x}}\kappa^4\sum_{t=1}^{T}\left(\sum_{k=0}^{\tau_t-1}\left\|\tilde{\nabla}_{\mathbf{x}}F_{t,w}(\mathbf{x}_t^k,\mathbf{y}_t^k) - \nabla_{\mathbf{x}}F_{t,w}(\mathbf{x}_t^k,\mathbf{y}_t^k)\right\|^2\right)$$

$$\leq \sum_{t=1}^{T}(\Phi_{t,w}\left(\mathbf{x}_t\right) - \Phi_{t,w}\left(\mathbf{x}_{t+1}\right)) + 1152\eta_{\mathbf{x}}\kappa^3\ell^2\sum_{t=1}^{T}\delta_{t,w}^0$$

$$\leq \frac{1}{w}\sum_{t=1}^{T}\left(f_t(\mathbf{x}_t,\mathbf{y}_{t,w}^*(\mathbf{x}_t)) - f_{t-w}(\mathbf{x}_t,\mathbf{y}_{t,w}^*(\mathbf{x}_t))\right) - \Phi_{T+1,w}(\mathbf{x}_{T+1}) + 1152\eta_{\mathbf{x}}\kappa^3\ell^2\left\{\frac{(T-1)\delta^2}{4\ell^2w^2}\right.$$

$$+ \frac{2}{\mu^2(w-1)^2}\sum_{t=2}^{T}\|\nabla_{\mathbf{y}}f_t(\mathbf{x}_{t-1}^{\tau_{t-1}},\mathbf{y}_{t,w}^*(\mathbf{x}_{t-1}^{\tau_{t-1}})) - \nabla_{\mathbf{y}}f_{t-w}(\mathbf{x}_{t-1}^{\tau_{t-1}},\mathbf{y}_{t-1,w}^*(\mathbf{x}_{t-1}^{\tau_{t-1}}))\|^2 + D^2\right\}$$

$$\leq \frac{2TM}{w} + M + \frac{288T\eta_{\mathbf{x}}\kappa^3\delta^2}{w^2} + \frac{2304\eta_{\mathbf{x}}\kappa^3\ell^2}{\mu^2(w-1)^2}V_w^2[T] + 1152\eta_{\mathbf{x}}\kappa^3\ell^2D^2,$$

where the first inequality follows from Assumption [4].

Thus

$$\tau \leq \frac{1}{\eta_{\mathbf{x}}}\frac{2MTw + \frac{9\delta^2T}{\ell} + \frac{72\ell w^2}{\mu^2(w-1)^2}V_w^2[T] + w^2M + \frac{5\ell D^2w^2}{32}}{\left(\frac{\delta^2}{3} - 2688\kappa^4\sigma^2\right)}.$$

$\square$

