# OpenReview forum: "Online Min-max Problems with Non-convexity and Non-stationarity"
_TMLR — Accepted by TMLR_

### Review · Reviewer_bH6d · 2023-04-09

**Summary Of Contributions:**

The paper considered the problem of online mix-max optimization, expressed as $\min_x \max_y f_t(x,y)$. The paper differs from existing works in two important aspects:
- firstly, the function $f_t$ is time-varying, reflecting the changing nature of non-stationary environments;
- secondly, $f_t$ is non-convex in $x$ and strongly concave in $y$, which presents unique challenges in solving the problem.

To address this problem, the authors introduce a new performance measure, referred to as the local NE-regret, as well as two non-stationarity measures for the variation of gradients. The authors propose an algorithm called SODA, which guarantees $O(T/w^2)$ local NE-regret with $O(T w)$ gradient oracle complexity, where $w$ represents the size of the window, a hyper-parameter in the algorithm. Finally, the authors extend their results to stochastic first-order feedback.

**Audience:**

Yes

**Broader Impact Concerns:**

The work is mostly theoretical and the broader impact discussion is not applicable.

**Claims And Evidence:**

Yes

**Requested Changes:**

There are several major concerns that should be resolved before acceptance. Other minor comments would be also helpful for improving the paper.

1.	(Major) The authors should provide more intuition and motivation for the artificial definitions and formulations to make this work more convincing to readers. This would require a careful examination of how these definitions and formulations relate to real-world applications.
2.	(Major) It is necessary that the authors could discuss the difference between this work and the [Hazan et al., 2017] in terms of techniques. Are there some fundamental difficulties in extending the results of [Hazan et al., 2017]?
3.	(Minor) Are there some unique challenges in extending the results to the stochastic first-order oracle?
4.	(Minor) In Section 1.1, when illustrating the main contributions of this work, $w$ is not defined yet.
5.	(Minor) At the beginning of Section 1.2: $f_t(x,y) = x^T A_t y$, but not $f_t(x,y) = x^T A_t y_t$.
6.	(Minor) The line before Definition 1: a missing space before ‘Thus’.
7.	(Minor) In my opinion, removing the definition of the NE-regret of [Rivera et al., 2018] and the dynamic NE-regret of [Zhang et al., 2022] may be better. The authors could directly discuss the reason for choosing the gradient norm as the performance and why it can capture the dynamic nature. The current statements are a little confusing.
8.	(Minor) The authors could consider changing the abbreviation of their algorithm from "SODA" to avoid confusion with the top TCS conference of the same name.


**Strengths And Weaknesses:**

### Strengths
One of the strengths of this work is the significance of the problem it addresses, as non-convexity and non-stationary environments are commonly encountered in real-world applications. The authors' proposed solution for the problem is complete. Additionally, the paper is well-written, with clear explanations of the problem, its motivation, the proposed solution, and its extension.

### Weaknesses
There are several weaknesses in this work that should be addressed.
1. The function $f_t$ with non-convex $x$ and strongly concave $y$ seems somewhat artificial. While the authors argue that this kind of problem can arise in time-varying two-player zero-sum stochastic games, where the policies are modeled by DNNs with strong regularization, it is not clear whether this is a common problem in real-world applications.
2. The parameter $x$ is unbounded while $y$ is bounded, and the intuition behind this choice is not fully explained.
3. The definitions of the non-stationarity measures $V_{x,w}[T]$ and $V_{y,w}[T]$ lack intuition. It is not clear why the gradient of $f_{t-w}$ is measured by $y_{t,w}^*(x)$ with a mismatch in the time index, nor is it clear why the analogous quantity in $V_{y,w}[T]$ is in terms of $y_{t-1,w}^*(x)$ but not $y_{t,w}^*(x)$ as in $V_{x,w}[T]$.
4. The algorithm shares high similarity with [Hazan et al., 2017] (ICML 2017) Efficient Regret Minimization in Non-Convex Games, but without necessary comparison and discussion, where [Hazan et al., 2017] mainly focuses on the standard online learning setting with only one player.
5. The extension to the stochastic first-order oracle also appears in the [Hazan et al., 2017].
Overall, addressing these weaknesses could improve the clarity and impact of the paper.

---

> ### Author Response · Authors · 2023-05-11
> **Response to Reviewer bH6d**
>
> We thank the reviewer very much for your effort to provide a thorough and constructive feedback. We next address your comments.
>
> **Q.** The function with non-convex $x$ and strongly concave $y$ seems somewhat artificial.
>
> **A.** This comment is also given by the reviewer  mpMY. Based on the reviewer mpMY's suggestion, we have added a motivating example about time-varying GANs in Section 2 to demonstrate the relevance and importance of our online nonconvex-concave setting in real-world scenarios. Specifically, we show that our online nonconvex-concave framework is well-suited to model the training of a time-varying GAN, where a time-varying objective function can naturally be nonconvex with respect to the generator's parameters and (strongly) concave with respect to the discriminator's parameters.
>
> **Q.** The parameter $x$ is unbounded while $y$ is bounded,  and the intuition behind this choice is not fully explained.
>
> **A.** We note that this choice of unbounded $x$ and bounded $y$ is commonly used in the existing studies of nonconvex-concave min-max optimization [1][2][3]. Moreover, in our work, we have made this assumption for technical convenience, as explained in Section 6.2, where we need to control a quantity $\delta_{t,w}^{0}=||y_{t,w}^{*}(x_{t}^{0})-y^{0}_t||^2$. We further explain that this quantity at $t=1$ is bounded by the diameter $D$ of the $\mathcal{Y}$-space, which requires $\mathcal{Y}$ to be bounded.
>
> To clarify this choice, we have added an explanation in the relevant section of the paper, along with references to previous studies where similar assumptions have been made.
>
> **Q.** The definitions of the non-stationarity measures
>
> **A.** Thanks for your feedback. We have made slight modifications to the notions of non-stationarity measures in Definition 2. Additionally, we have included a remark below Definition 2 to provide further clarification and intuition behind these measures.
>
> **Q.** Comparison with [Hazan et al., 2017]
>
> **A.** The online nonconvex setting studied in [Hazan et al., 2017] is a special case of our work (if we view $\mathcal{Y}$ as a singleton). We have highlighted connections between our work with [Hazan et al., 2017] in several parts of the paper, including the paragraphs in Section 3.1 titled ``Why Sliding-window Averaging?``, ``Discussions about Stopping Criterion``, ``Robustness of TSODA``, ``Optimality of Regret Bound``. Furthermore, in Section 6, we have added a paragraph titled ``Significance of Techniques`` to further summarize the differences between our work and the approach taken in [Hazan et al., 2017], particularly in terms of the techniques.
>
> **Q.** Are there some unique challenges in extending the results to the stochastic first-order oracle?
>
> **A.** Yes. Similarly to the challenges of SGD w.r.t. GD, when extending the results to the stochastic first-order oracle (SFO), SFO is not the true gradient of the loss function, but a stochastic approximation of the true gradient. As a result, in Algorithm 2, when we use SFO to perform gradient descent ascent update, errors due to such randomness causes the additional in the convergence of the gradient descent ascent, which will require the assumptions of unbiasedness and bounded variance (Definition 3 and Assumption 4).
>
> **Q.** When illustrating the main contributions of this work, $w$ is not defined yet.
>
> **A.** We added a definition for $w$ in the contribution.
>
> **Q.** At the beginning of Section 1.2: $f_t(x, y)=x^T A_t y$, but not $f_t(x, y)=x^T A_t y_t$.
>
> **A.** We fixed the typo.
>
> **Q.** The line before Definition 1: a missing space before ‘Thus’.
>
> **A.** Fixed.
>
> **Q.** In my opinion, removing the definition of the NE-regret of [Rivera et al., 2018] and the dynamic NE-regret of [Zhang et al., 2022] may be better.
>
> **A.** Thanks for the suggestion. We have rewritten the paragraph titled ``Why Norm of Gradient as Metric?`` for clarity.
>
> **Q.** The authors could consider changing the abbreviation of their algorithm from "SODA" to avoid confusion with the top TCS conference of the same name.
>
> **A.** Thanks for your feedback. Changed "SODA" to "TSODA".
>
> [1] (Yang et al, 2022) Nest Your Adaptive Algorithm for Parameter-Agnostic Nonconvex Minimax Optimization.
>
> [2] (Li et al, 2022) TiAda: A Time-scale Adaptive Algorithm for Nonconvex Minimax Optimization.
>
> [3] (Lin et al, 2019) On Gradient Descent Ascent for Nonconvex-Concave
> Minimax Problems.

---

### Review · Reviewer_mpMY · 2023-04-17

**Summary Of Contributions:**


This paper studies an online minimax optimization problem, where the objective function is nonconvex in $x$ but strongly concave in $y$. The authors propose a new performance metric, local Nash equilibrium (NE)-regret, which extends the commonly used metric for offline nonconvex-strongly-concave minimax optimization. The authors also define total-variation-type measures to quantify the variability of the environment within a time window. On top of these settings, the authors propose the (deterministic) SODA algorithm, which is OGDA but using the time-window average $F_{t, w}$ as the objective function. Then, the authors establish theoretical guarantees, that is, bounds on the local NE regret and the total number of iterations, where the latter is proportional to the total number of gradient oracle calls. Furthermore, the authors consider the setting of a stochastic first-order oracle (being unbiased & having bounded variance) instead of having full function gradient access. For this setting, they propose a variant of the algorithm, SODA-SFO, and establish its tractability, regret guarantee, and iteration bound.


**Audience:**

Yes

**Broader Impact Concerns:**

N.A.

**Claims And Evidence:**

Yes

**Requested Changes:**

Definition 2 & Remark 1:
Note that the first few terms in (5) won’t be 0 under the periodicity assumption. Therefore, $V_{x,w}[T] = V_{y,w}[T] = 0$ is false, unless you define $f_{-1}$, $f_{-2}$ etc. as having periodical values instead of 0. But these are currently defined as $0$ in Section 3.1 and this is used throughout. If you keep the definition of $f_{-1} = f_{-2} = \dots = 0$, then under periodicity, the variation would likely be $O(w)$, which is ok if period length $ = w \ll T$.

In proof of Lemma A.3:
Can you elaborate the use of the global error bound condition in Drusvyatskiy & Lewis (2018)? I think the RHS of the last inequality in the proof of A.3 is wrong. It looks like a copy-paste error.
It should instead be the first term in the RHS of the first inequality in this proof, aka norm of the prox-gradient (projected gradient) mapping. In addition, please further elaborate the use of the global error bound condition in the proof as it’s a key result for this lemma and hence for this work. Corollary 3.6 in D&L (2018) seems to be the right result you are using. You might want to restate it in your notation as a separate lemma. The deduction should be: strong convexity (concavity) of $F_{t,w}(x, \cdot)$ ⇒ quadratic growth ⇒ error bound condition with RHS being the norm of the prox-gradient mapping.

The term "regret" suggests a comparison with a better objective value in hindsight from an optimal policy. I would really name $R_{w-NE}$ as "aggregate time-window averaged leader gradient residuals". You can keep the existing name, but may need to explain how it's indeed a "regret"-type metric. Maybe: In the special convex-concave case, a lower $w$-local NE regret (defined in this paper) implies a lower NE regret (defined in Rivera et al., 2018).

Writing suggestions (take them at your discretion)

Maybe: $R_{w-NE}$ → $R^{\rm NE}_{w}$

In the paragraph starting with "Why sliding window averaging?": … a typical notion that people are interested in real-world applications (there might need to be another “in” after “interested”; in general I would avoid phrases like “people are interested in”; try using “a typical/intuitive notion commonly used in practice”)

Section 3.2 Remark 1: For the well-timed $w$ (“the” → “a”)

Section 3.2 Remark 1: could be considerably small (small → smaller)

Proof of Lemma A.1: Since the averaging maintains the strongly-nonconcavity and smoothness” (no “the”)

Proof of Lemma A.2: strongly-convave (typo)

Proof of Lemma A.2: in term of notions about $\nabla F_{t,w}$ (remove “notions about”)

Section 5.2:
You may want to start this section by “Given the SFO and the inner-loop termination condition in line 6 of Algorithm 2, one immediate question is whether Algorithm 2 terminates in finite time. To this end, we first establish that…”; Also computation tractable → computationally tractable; you may simply remove the last clause “which justifies that …”.

 __expectation__ local NE-regret vs. __expected__ local NE-regret?

Theorem 3: $\tau_t$ and $\tau$ is → $\tau_t$ and $\tau$ are

Paragraph under Theorem 4:
“Beyond finite stopping…” Fix this sentence. Maybe start this paragraph as “Beyond …, we next provide an upper bound on the overall iteration complexity of SODA-SFO. Similar to Li & Orabona … , we adopt the following stronger boundedness assumption on the SFO.”


**Strengths And Weaknesses:**

__Strength__

Overall correct result statements and proofs. Clear structure and writing. Clear assumption and result statements. Adequate justifications for choosing the local NE regret with a preset time window over the existing NE regret notion in online convex-concave minimax problems. Checked all proofs and did not find severe flaws.

__Weaknesses__

One main point to address is whether the nonconvex-strongly-concave setting is indeed practically motivated. It would be good to have a few simple, structured, practically motivated examples (or references that suggest these) of decision-making problems that are not well captured by existing settings but fit the new nonconvex-strongly-concave setting of this paper. Arguably, "online deep learning" is very broad and breaks every assumption. They don't have to satisfy all assumptions but should really demonstrate the value/meaningfulness of the nonconvex-strongly-concave setting. For example, some previous papers (on different problem settings) have included such brief discussions:
- Immorlica et al. (https://arxiv.org/pdf/1811.11881.pdf) has a dynamic pricing example and a pointer to Badanidiyuru et al. (2018) for their BwK problem setting.
- Agarwal et al. (http://proceedings.mlr.press/v99/agarwal19a/agarwal19a.pdf) discusses the implication of their results to non-convex games and, in particular, GANs. They also emphasized the relevance of the more broader area of online learning in games at the very beginning.

See Requested Changes for further questions.

---

> ### Author Response · Authors · 2023-05-11
> **Response to Reviewer mpMY**
>
> We thank the reviewer very much for your effort to provide a thorough and constructive feedback. We next address your comments.
>
> **Q.** One main point to address is whether the nonconvex-strongly-concave setting is indeed practically motivated.
>
> **A.** Thanks for providing us with references about how existing works discuss practically motivated examples. We have taken your suggestions into consideration and added a motivating example about
> time-varying GANs in Section 2 to demonstrate the relevance and importance of our work in real-world scenarios.
>
> **Q.** Definition 2 \& Remark 1: $V_{x,w}[T]=V_{y,w}[T]=0$ is false.
>
> **A.** Indeed, in the periodic setting, the variation  would likely be $O(w)$. It has been fixed in the revision.
>
> **Q.** Comments on the proof of Lemma A.3
>
> **A.** The RHS of the last inequality in the proof of A.3 is indeed a copy-paste error, and we have corrected it accordingly. Moreover, we have included the result of Corollary 3.6 in Drusvyatskiy \& Lewis (2018) in Lemma A.4 for completeness, along with a detailed explanation of how to use this result in the proof of Lemma A.3.
>
> **Q.** The term "regret" suggests a comparison with a better objective value in hindsight from an optimal policy.
>
> **A.** Thanks for the suggestions. In Section 3.1,  the paragraph  of ``Why Capturing the Dynamic Nature?``,  we have discussed the connection between our local NE-regret and the standard type regret for convex-concave games, which implicitly reveals that a lower local NE regret implies a lower dynamic NE regret in convex-concave setting. To demonstrate that local NE regret is indeed a ``regret"-type metric, we have added such implication at the end of this paragraph for clarity.
>
> **Q.** Writing suggestions.
>
> **A.** Many thanks for your suggestions. We have made the changes based on your suggestions in the revision.

---

### Review · Reviewer_zzmX · 2023-04-29

**Summary Of Contributions:**

This paper studies online nonconvex-concave min-max optimization problem. Specifically, its contributions are threefold:

1) This paper has introduced a new notion of regret, which is referred to as "local Nash equilibrium regret".

2) This paper has developed an algorithm, referred to as SODA; it has also developed its stochastic version (SODA-SFO)

3) This paper has analyzed SODA and SODA-SFO. In particular, it has provided both regret bounds (Theorem 1, Theorem 4) and complexity guarantees (Theorem 2, Theorem 3, Theorem 5).

**Audience:**

Yes

**Claims And Evidence:**

Yes

**Requested Changes:**

**1) About the "local Nash equilibrium regret":** My main concern is that the definition of the "local Nash equilibrium regret" might be problematic. In particular, I think it has the following issues: first, even if the Nash equilibrium (NE) regret developed in (Rivera et al. 2018) might be problematic. In particular, if the x-learner and the y-learner cooperate, they can achieve small NE-regret at (x, y) that is far away from the min-max optimal solution. However, this definition does not prevent the x-learner and the y-learner from cooperation. This seems to be problematic. Moreover, this NE-regret measures the **performance distance** (due to the absolute value), rather than the **performance loss**. This also seems to be problematic. I am wondering if this metric can be cheated by an algorithm in some practical problems?

The "local Nash equilibrium (NE) regret" seems to be more problematic. Specifically, as its name indicates, it is based on the magnitudes of the gradients (local information), rather than performance loss or distance. Intuitively, it measures how close the solution is to a stationary point (even a very bad stationary point). I am also wondering if this metric can be cheated by an algorithm in some problems?

I fully understand that the NE regret was developed in previous literature and the authors are following existing work. But still, I hope the authors can further clarify these issues in a revision since the regret definition is crucial for this paper.

**2) Experiment results:** I recommend the authors to add some experiment results to the paper. In particular, the proposed algorithms take several parameters, including the window size $w$, the step sizes $\eta_x$ and $\eta_y$, and the tolerance $\delta$. Theoretical results further show that the optimal choices of the step sizes depend on parameters like $\ell$ and $\mu$, **which are typically unknown in practice**. Thus, I think it is necessary to demonstrate some experiment results to illustrate the robustness of the proposed algorithms to possible misspecification of these parameters.

I recommend the authors to run SODA and/or SODA-SFO on one or two practical problems and sweep over $w$ and the step sizes. The experiment results should demonstrate if the proposed algorithms are robust to possible misspecification of the algorithm parameters.

**3) Writing:** Overall, the writing of this paper is pretty good. However, I recommend the authors to add proof sketches for key results (e.g. Theorem 1) to the main body of the paper so that the readers can get the high-level idea of the analysis without diving into the appendices.






**Strengths And Weaknesses:**

**Strengths:**

- The considered problem is interesting.

- To the best of my knowledge, the analyses in this paper are correct.

**Weaknesses:**

- My understanding is that the definition of the "local Nash equilibrium regret" might be problematic, which I will discuss in detail below.

- This paper does not have any experiment results, which might be crucial to understand the robustness of the proposed algorithms to possible parameter misspecification.

- The writing of this paper can be further improved.

---

> ### Author Response · Authors · 2023-05-11
> **Response to Reviewer zzmX**
>
> We thank the reviewer very much for your effort to provide a thorough and constructive feedback. We next address your comments.
>
> **Q.** On the "local Nash equilibrium regret":
>
> *(a)* If x-learner and the y-learner cooperate, they can achieve small NE-regret at $(x, y)$ that is far away from the min-max optimal solution.
>
> *(b)* NE-regret measures the performance distance (due to the absolute value), rather than the performance loss.
>
> *(c)* "Local Nash equilibrium (NE) regret" measures how close the solution is to a stationary point (even a very bad stationary point).
>
> **A.**
>
> *(a)* We agree with the reviewer's comment. However, our goal here is to find the min-max optimal solution defined as $\min_x f(x,y^*(x))$, where $y^*(x):= \text{argmax}_y f(x,y)$. Here, $(x,y)$ cannot take any arbitrary values to reach the minimal value of $f(x,y)$, but have to be in the form of $(x,y^*(x))$. In this sense, x-learner and the y-learner cannot fully cooperate, but have to follow the specific way of $(x,y^*(x))$ to minimize the function $f(x,y)$.
>
> *(b)* The definition of regret in terms of distance aims to capture how the cumulative loss (or gradient norm) achieved by having only sequential information is different from that by having the hindsight information. Namely, such a metric captures the impact of sequential information (which is the unique characteristic of online learning) on the learning performance.
>
> *(c)* Using "local NE regret" is due to the nonconvexity of the function with respect to $x$. Such a problem has been shown to be NP hard, and "local NE regret" (or "local NE" in the offline case) serves as a standard surrogate metric, which are feasible for polynomially efficient algorithm to attain. This is also the reason that achieving a stationary point has been widely adopted as a feasible convergence metric in offline nonconvex optimization.
>
> **Q.** Experiment results.
>
> **A.** Thanks for the suggestions. We conducted experiments on the widely-used Wasserstein robustness model for TSODA.  We investigated the effect of window size on  the performance and compared its effectiveness with other baseline algorithms. The experimental setup, results, and discussions have been added in a new Section 7 in the revision.
>
> Regarding the effect of stepsize, it is worth noting that the performance of many optimization methods can be sensitive to changes in stepsize. As a result, it is common practice to tune the stepsize through a trial-and-error process to achieve optimal performance, rather than selecting it based on prior knowledge of Lipschitz parameters. We have taken this into consideration and ensured that our results are robust to reasonable variations in stepsize.
>
> We would also like to mention that there have been recent efforts in nonconvex min-max optimization, e.g. [1][2], towards developing adaptive algorithms, which adjust their stepsize and search directions based on past gradients. We acknowledge that stepsize selection is an important aspect of optimization, and we will continue to explore adaptive algorithms in our future work.
>
> **Q.** Writing. I recommend the authors to add proof sketches for key results (e.g. Theorem 1) to the main body of the paper.
>
> **A.** Thanks for your suggestions. We have added a new Section 6 in the revision, which provides the proof outlines for Theorems 1 and 2 and explains the key components essential for the proofs.
>
> [1] (Yang et al, 2022) Nest Your Adaptive Algorithm for Parameter-Agnostic Nonconvex Minimax Optimization.
>
> [2] (Li et al, 2022) TiAda: A Time-scale Adaptive Algorithm for Nonconvex Minimax Optimization.

---

### Author Response · Authors · 2023-05-11
**Revision Uploaded**

We have uploaded a revised paper, which incorporates the valuable suggestions provided by the reviewers. Below is a summary of main changes we have made, which are highlighted by the blue-colored texts in the revisions.

1. We have added a motivating example in Section 2, where we use an architecture of time-varying GANs to provide a practical justification for our online nonconvex-strongly-concave setting.
2. We have included Section 6 to present the outline of proof and clarify the new technical development of our work.
3. We have renamed our proposed algorithm from SODA to TSODA to avoid potential confusion with a top TCS conference.
4. We have evaluated TSODA on the widely-used Wasserstein robustness model focusing on the impact of the window size and comparison with some baseline algorithms.

---

### Decision · Action_Editors · 2023-06-07

**Recommendation:** Accept with minor revision

**Comment:**

In this paper, the authors consider online nonconvex-strongly-concave min-max optimization in the nonstationary environment. They use local Nash equilibrium (NE)-regret as the performance measure, and propose a new algorithm named as TSODA to achieve the optimal regret. Furthermore, the authors extend their result to the setting with stochastic first-order feedback. Some experiments are done to evaluate the performance of the proposed algorithm.

The paper makes significant contributions to online nonconvex-strongly-concave min-max optimization. After discussions, all the reviewers are generally positive with this paper. But there still exist several issues to be addressed:
(a) More justifications of the "local Nash equilibrium regret”;

(b) A more accurate title;

(c) More comparisons with Lin et al. (2020a).


**Audience:**

Yes

**Claims And Evidence:**

Yes